# TspanC8 tetraspanins differentially regulate ADAM10 endocytosis and half-life

Etienne Eschenbrenner[1,2], Stéphanie Jouannet[1,2], Denis Clay[2,3], Joëlle Chaker[1,2], Claude Boucheix[1,2], Christel Brou[4], Michael G Tomlinson[5], Stéphanie Charrin[1,2], Eric Rubinstein[1,2]

**ADAM10 is a transmembrane metalloprotease that is essential for development and tissue homeostasis. It cleaves the ectodomain of many proteins, including amyloid precursor protein, and plays an essential role in Notch signaling. ADAM10 associates with six members of the tetraspanin superfamily referred to as TspanC8 (Tspan5, Tspan10, Tspan14, Tspan15, Tspan17, and Tspan33), which regulate its exit from the endoplasmic reticulum and its substrate selectivity. We now show that ADAM10, Tspan5, and Tspan15 influence each other's expression level. Notably, ADAM10 undergoes faster endocytosis in the presence of Tspan5 than in the presence of Tspan15, and Tspan15 stabilizes ADAM10 at the cell surface yielding high expression levels. Reciprocally, ADAM10 stabilizes Tspan15 at the cell surface, indicating that it is the Tspan15/ADAM10 complex that is retained at the plasma membrane. Chimeric molecules indicate that the cytoplasmic domains of these tetraspanins contribute to their opposite action on ADAM10 trafficking and Notch signaling. In contrast, an unusual palmitoylation site at the end of Tspan15 C-terminus is dispensable. Together, these findings uncover a new level of ADAM10 regulation by TspanC8 tetraspanins.**

## Introduction

Many cell and developmental processes are regulated by a proteolytic cleavage of membrane-anchored proteins in their extracellular region, a process referred to as ectodomain shedding. Several proteases have been shown to be involved in this process, including several members of the ADAM (a disintegrin and metalloprotease domain) family of membrane-anchored metalloproteases (Blobel, 2005; Saftig & Reiss, 2011; Lichtenthaler et al, 2018). ADAM10 is one of the most extensively characterized ADAM proteases. It mediates the ectodomain shedding of dozens of transmembrane proteins, including

adhesion proteins such as E- and N-cadherins, growth factor precursors, and cytokines (Saftig & Reiss, 2011). ADAM10-mediated cleavage of the amyloid precursor protein prevents the formation of the amyloid peptide Aβ, a major component of amyloid plaques observed in Alzheimer's disease (Saftig & Lichtenthaler, 2015). ADAM10 also plays an essential role in Notch signaling. Binding of a Notch ligand to the receptor allows sequential cleavage by ADAM10 and the γ-secretase complex, resulting in the release of Notch intracellular domain and its translocation to the nucleus where it regulates the transcription of Notch target genes (Bozkulak & Weinmaster, 2009; Kopan & Ilagan, 2009; van Tetering et al, 2009; Groot et al, 2014). Importantly, ADAM10-deficient mice die during development, and its tissue-specific ablation yields abnormalities in various organs that are associated with a defect in Notch signaling (Saftig & Lichtenthaler, 2015; Dempsey, 2017; Alabi et al, 2018; Lambrecht et al, 2018).

The activity of ADAM10 is regulated by both intrinsic properties and extrinsic factors. ADAM metalloproteases are synthesized as zymogens that remain catalytically inactive until the prodomain is released after cleavage by pro-protein convertases during transport to the cell surface (Blobel, 2005; Saftig & Reiss, 2011; Lichtenthaler et al, 2018). The recent crystal structure of the entire ADAM10 ectodomain revealed that the disintegrin and cysteine-rich domains envelope the metalloproteinase domain, concealing the active site of the enzyme and probably restricting substrate access and preventing broad-spectrum activity of the mature protease at the cell surface (Seegar et al, 2017). In addition, ADAM10 activity and substrate selectivity is regulated by a number of interacting proteins (Vincent, 2016), including several members of the tetraspanin superfamily.

Tetraspanins are expressed by all metazoans and are characterized by four transmembrane domains that flank two extracellular domains of unequal size, conserved key residues, and a specific fold of the large extracellular domain. Genetic studies in humans or mice have shown their key role in a number of physiological processes, including immunity, vision, kidney function,

---

[1]Inserm, U935, Villejuif, France   [2]Université Paris-Sud, Institut André Lwoff, Villejuif, France   [3]Inserm, Unité Mixte de Service UMS33, Villejuif, France   [4]Institut Pasteur, Unit of Membrane Trafficking and Pathogenesis, Department of Cell Biology and Infection, Paris, France   [5]School of Biosciences, College of Life and Environmental Sciences, University of Birmingham, Birmingham, UK

Correspondence: eric.rubinstein@inserm.fr
 Stéphanie Charrin and Eric Rubinstein's present address is Sorbonne Université, INSERM, CNRS, Centre d'Immunologie et des Maladies Infectieuses, CIMI-Paris, Paris, France

reproduction, muscle regeneration, and mental capacity (Hemler, 2003; Charrin et al, 2009, 2014). A major feature of these molecules is their association with many other integral proteins, thus building a dynamic network of interactions referred to as the "tetraspanin web" or tetraspanin-enriched microdomains (Hemler, 2003; Charrin et al, 2009, 2014). Inside this network, tetraspanins interact directly with a limited number of partner proteins to form primary complexes which associate with one another. We and others have recently demonstrated that ADAM10 has six tetraspanin partners (Tspan5, Tspan10, Tspan14, Tspan15, Tspan17, and Tspan33) which mediate its exit from the ER and belong to a subgroup of tetraspanins having eight cysteines in the largest of the two extracellular domains and referred to as TspanC8 (Dornier et al, 2012; Haining et al, 2012; Prox et al, 2012). The regulation of ADAM10 trafficking by TspanC8 tetraspanins is evolutionary conserved because Tsp-12 in *Caenorhabditis elegans* and the three Drosophila TspanC8 tetraspanins regulate ADAM10 subcellular localization in vivo (Dornier et al, 2012; Wang et al, 2017).

The regulation of ADAM10 by tetraspanins has important consequences for Notch signaling. Mutations of the TspanC8 tetraspanin Tsp-12 in *C. elegans* genetically interacted with Notch or ADAM10 mutations (Dunn et al, 2010). In addition, depletion of the three Drosophila TspanC8 tetraspanins impaired several Notch-dependent developmental processes and Notch activity in vivo (Dornier et al, 2012). Importantly, depletion of only one or two of the Drosophila TspanC8 produced only mild defects, suggesting that these tetraspanins compensate for one another. In mammals, Tspan5 and Tspan14, which are with Tspan17 the more closely related to Tsp-12 and Drosophila TspanC8, were also shown to be positive regulators of Notch signaling (Dornier et al, 2012). In contrast, Tspan15 was shown to be a negative regulator (Jouannet et al, 2016) and the effect of Tspan33 varied among studies (Dunn et al, 2010; Jouannet et al, 2016; Ruiz-Garcia et al, 2016). More generally, TspanC8 tetraspanins have a different impact on the cleavage of various ADAM10 substrates (Matthews et al, 2017; Saint-Pol et al, 2017b). Notably, only Tspan15 seems to be required for ADAM10-mediated N-cadherin shedding (Prox et al, 2012; Jouannet et al, 2016; Noy et al, 2016; Seipold et al, 2017).

Comparing the properties of ADAM10 differentially regulated by different TspanC8 tetraspanins will give clues on how tetraspanins regulate ADAM10 substrate selectivity. Using ADAM10 deletion mutants and ADAM10 chimeras with ADAM17, the most closely related ADAM to ADAM10, Noy et al (2016) have shown that TspanC8 engage ADAM10 in subtly different ways, which led the authors suggest that they might direct substrate specificity by constraining ADAM10 into defined conformations (Noy et al, 2016). We have also shown that Tspan5 and Tspan15 influence ADAM10 membrane compartmentalization, with Tspan5 promoting and Tspan15 restricting ADAM10 interaction with the tetraspanin web (i.e., with other tetraspanins and tetraspanin-associated molecules) (Jouannet et al, 2016), suggesting that the interaction of ADAM10 with the tetraspanin web may favor the cleavage of discrete substrates and prevents the cleavage of others. We now demonstrate that although both Tspan5 and Tspan15 target ADAM10 to the plasma membrane, they have an opposite effect on ADAM10 endocytosis and turnover.

**Table 1. Expression of Tspan5, Tspan14, and Tspan15 in the three cell lines used in the study and consequence of the manipulation of Tspan5 and Tspan15 expression levels on ADAM10 surface expression and endocytosis.**

| | PC3 | U2OS | HeLa |
|---|---|---|---|
| Surface expression in WT cell lines | | | |
| ADAM10 | ++ | ++ | + |
| Tspan5 | + | + | 0 |
| Tspan15 | ++ | 0 | 0 |
| Tspan14 | 0 | ++ | + |
| ADAM10 surface expression | | | |
| Tspan5 transfection | ↓ | = | ↑ |
| Tspan15 transfection | = | ↑↑ | ↑ |
| KO Tspan5 | = | = | NA |
| KO Tspan15 | ↓ | NA | NA |
| ADAM10 endocytosis | | | |
| Tspan5 transfection | ↑ | ↑ | = |
| Tspan15 transfection | = | ↓ | ↓ |
| KO Tspan5 | = | ↓ | NA |
| KO Tspan15 | ↑ | NA | NA |

The expression level of Tspan14 is only determined at the RNA level. NA, not applicable.

# Results

As evidenced in the current study, the result of manipulating the expression level of a given TspanC8 tetraspanin depends on the TspanC8 repertoire. This is why we use three different cellular models of known TspanC8 repertoires. U2OS cells express Tspan5 and Tspan14, PC3 cells express Tspan5 and Tspan15, and HeLa cells express only a low level of Tspan14, and this low overall expression level of TspanC8 tetraspanins results in a substantial retention of ADAM10 in the ER (Dornier et al, 2012; Jouannet et al, 2016) (Table 1).

### Evidence that Tspan5 and Tspan15 have a different impact on ADAM10 trafficking, half-life, and molecular interactions

We have previously demonstrated that transfection of GFP-tagged Tspan5 and Tspan15 in U2OS cells induces an increase in ADAM10 surface expression levels of three and five times, respectively (Jouannet et al, 2016). Surprisingly, expression of untagged Tspan5 in a Tspan5/Tspan15–positive cell line (PC3) led to a 40% reduction of ADAM10 surface levels, whereas overexpression of Tspan15 had no effect (Fig 1A and Table 1). There was no change in ADAM10 gene expression as determined by RT-qPCR (data not shown). We were concerned that the GFP tag could modify to some extent of the properties of these tetraspanins and transfected the untagged Tspan5 and Tspan15 in U2OS cells, which express Tspan5 and Tspan14 but not Tspan15. Both tetraspanins were expressed at the cell surface and in an intracellular compartment (Figs 1B and S1A). ADAM10 colocalized with Tspan5 but not with Tspan15 inside the cells (Fig S1B). Contrasting to our previous observations using GFP-tagged

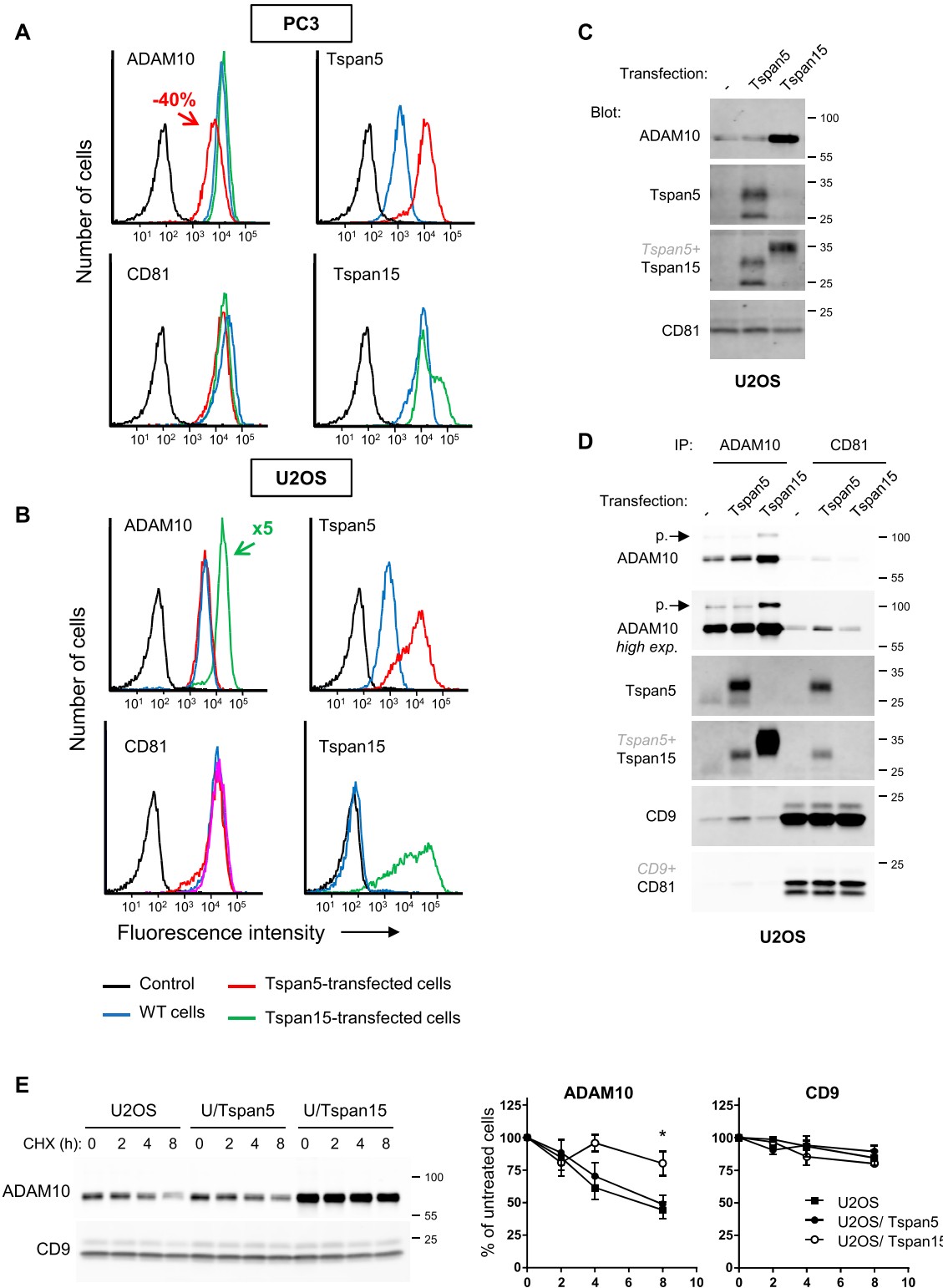

**Figure 1. Opposite effect of Tspan5 and Tspan15 on ADAM10 expression level and turnover.**
**(A, B)** Flow cytometry analysis of the surface expression of ADAM10, Tspan5, Tspan15, and CD81 in PC3 (A) or U2OS (B) cells transfected or not with Tspan5 or Tspan15. **(C)** Western blot analysis of ADAM10, Tspan5, Tspan15, and CD81 expression in U2OS cells transfected or not with Tspan5 or Tspan15. **(D)** U2OS cells transfected or not with Tspan5 or Tspan15 were lysed in the presence of Brij 97 and divalent cations to preserve tetraspanin to tetraspanin interactions, before immunoprecipitation of ADAM10 or CD81. The composition of the complexes was analyzed by Western blot. The proform (p.) of ADAM10 is indicated by an arrow. **(E)** U2OS cells transfected or not with Tspan5 or Tspan15 were incubated for the indicated time with 100 μg/ml cycloheximide (CHX) before lysis. For increased sensitivity, the amount of ADAM10 present in the

molecules, if untagged Tspan15 stimulated a large increase in ADAM10 surface levels (>5 times), untagged Tspan5 had little effect on ADAM10 surface levels (Fig 1B and Table 1). Western blot analysis showed that Tspan15, but not Tspan5, induced a large increase in total ADAM10 expression level in U2OS cells (Fig 1C). This increase in ADAM10 protein level was not associated with a change in mRNA levels (data not shown). Co-immunoprecipitation experiments showed that despite a higher expression level after Tspan15 transfection, ADAM10 was proportionally less associated with canonical tetraspanins such as CD9 and CD81, which is consistent with the lack of co-immunoprecipitation of Tspan15 with CD81 (Fig 1D). In contrast, Tspan5 was readily co-immunoprecipitated with CD81 and increased the association of ADAM10 with CD9 and CD81. Of note, ADAM10 is synthesized as a 100-kD zymogen that undergoes removal of the prodomain by pro-protein convertases in the Golgi to yield the mature 67-kD ADAM10 molecule. The expression of Tspan5 or Tspan15 did not reduce the level of ADAM10 proform, indicating that in U2OS cells, the amount of TspanC8 is not limiting for the exit of ADAM10 from the ER. In addition, in the ADAM10 IP, the apparent MW of the ectopically expressed Tspan5 is higher than that of endogenous Tspan5 (Fig 1D). This is most likely because the bulk of Tspan5 in transfected cells does not directly associate with ADAM10, which limits the glycosylation of the fraction of Tspan5 to which it directly associates (Saint-Pol et al, 2017a). This is substantiated by the finding that a large fraction of surface Tspan5 in U2OS/Tspan5 cells is recognized by TS5-1r, an antibody that only recognizes the free pool of Tspan5 (Fig S1D). Finally, Tspan5 was no longer detected in the ADAM10 IP in Tspan15-transfected cells, suggesting that ectopically expressed Tspan15 competes with Tspan5 for the interaction with ADAM10. Again, this is substantiated by an increase in the binding of TS5-1r (Fig S1D).

The large increase in ADAM10 levels observed after Tspan15 expression is due at least in part to the stabilization of ADAM10 molecules. Indeed, whereas the half-life of ADAM10 after blocking protein synthesis by cycloheximide was less than 8 h in U2OS or Tspan5-transfected cells, more than 80% of the initial pool was still present in Tspan15-transfected cells after 8 h of treatment (Fig 1E). CD9 decay was similar in parental and transfected U2OS cells, with more than 80% of CD9 remaining after 8 h of cycloheximide treatment.

## ADAM10 is variably internalized in different cell lines

The different impact of Tspan5 and Tspan15 on ADAM10 surface levels could be explained by a different residence time at the cell surface. To measure with minimal variations the rate of ADAM10 endocytosis, we incubated the cells for 1 h at 37°C with the anti-ADAM10 mAb 11G2 coupled to DyLight 650. After fixation or cooling to 4°C, the surface pool was then labelled with an anti-mouse polyclonal antibody coupled to another fluorochrome before confocal microscopy analysis. In the images, the internalized fraction not labelled by the secondary antibody appears in green, whereas the surface fraction appears in yellow/orange (see Fig 2B for a first example). The surface labelling is used to generate a mask in Icy imaging software, which

allows the surface pool to be removed from the DyLight 650 image and quantification of the endocytosed fraction. The fraction of ADAM10 internalized after 1 h incubation at 37°C varied according to the cell line studied ranging from 3% in PC3 cells to 18% in U2OS cells (Fig 2A). It was notably lower in cells expressing Tspan15 (PC3 and A549) than in the three other cell lines tested (U2OS, HeLa, and HCT116). In U2OS cells, the endocytosed fraction of ADAM10 was targeted to CD63-positive compartments, which include late endosomes and lysosomes (Fig 2C). A comparison with other surface molecules showed that the internalization rate of ADAM10 in U2OS cells was similar to that of CD81 and intermediate between that of the tetraspanins CD9 and CD63 (data not shown), the latter being known to undergo fast endocytosis at the plasma membrane (Pols & Klumperman, 2009).

## Tspan5 and Tspan15 differentially regulate ADAM10 endocytosis in PC3 and U2OS cells

We then analyzed whether Tspan5 and Tspan15 could modulate ADAM10 endocytosis. As shown in Fig 2B, transfection of Tspan5 in U2OS cells stimulated ADAM10 endocytosis, whereas very little endocytosed ADAM10 could be observed in U2OS/Tspan15 cells (Fig 2B and Table 1). In addition, expression of Tspan5 in PC3 cells stimulated ADAM10 endocytosis twofold, whereas overexpression of Tspan15 had no significant effect (Fig S2 and Table 1).

Antibodies to the ectodomain of proteins that traffic through the plasma membrane can bind to their target, even if the passage at the plasma membrane is transient (Lippincott-Schwartz & Fambrough, 1986) and are internalized with their target molecule. Thus, if ADAM10 associated with Tspan5 (ADAM10$_{Tspan5}$) undergoes faster internalization than ADAM10$_{Tspan15}$ after reaching the plasma membrane, an anti-ADAM10 mAb added to the medium at 37°C should accumulate more rapidly inside the cells in Tspan5-positive/Tspan15-negative cells. We incubated U2OS cells at 37°C with the DyLight 650–labelled ADAM10 mAbs for different times. After detachment, the cells were incubated with a PE-labelled secondary reagent, and the fluorescence was measured by flow-cytometry. DyLight 650 fluorescence measures the total pool of ADAM10 having passed through the plasma membrane (either still present or endocytosed) and the PE fluorescence the surface ADAM10 fraction. As shown in Fig 2D (left), the DyLight 650 fluorescence of parental U2OS cells or cells transfected with Tspan5 and incubated at 37°C with the ADAM10 mAb increased continuously over time, reaching after 4 h of incubation a level four times higher than that observed after labelling the cells at 4°C. Because there was no concomitant increase in ADAM10 surface levels (but instead a decrease, Fig 2D middle), the vast majority of this signal was intracellular and corresponded to internalized mAbs (Fig 2D, right). In contrast, the fluorescence of Tspan15-transfected cells was quite stable, indicating that a minor fraction of ADAM10 was internalized in these cells. After 20 h of incubation, the DyLight 650 staining of parental U2OS cells or Tspan5-transfected cells was of the same magnitude as that of Tspan15-transfected cells (Fig 2E),

lysates was determined after immunoprecipitation. The level of CD9 was determined by Western blotting on the lysates. On the right is shown a quantification of the levels of ADAM10 and CD9 according to the time of treatment. Data were statistically analyzed by a one-way ANOVA followed by a Dunnett's multiple comparisons test (*$P$ < 0.05, compared with WT, n = 4).

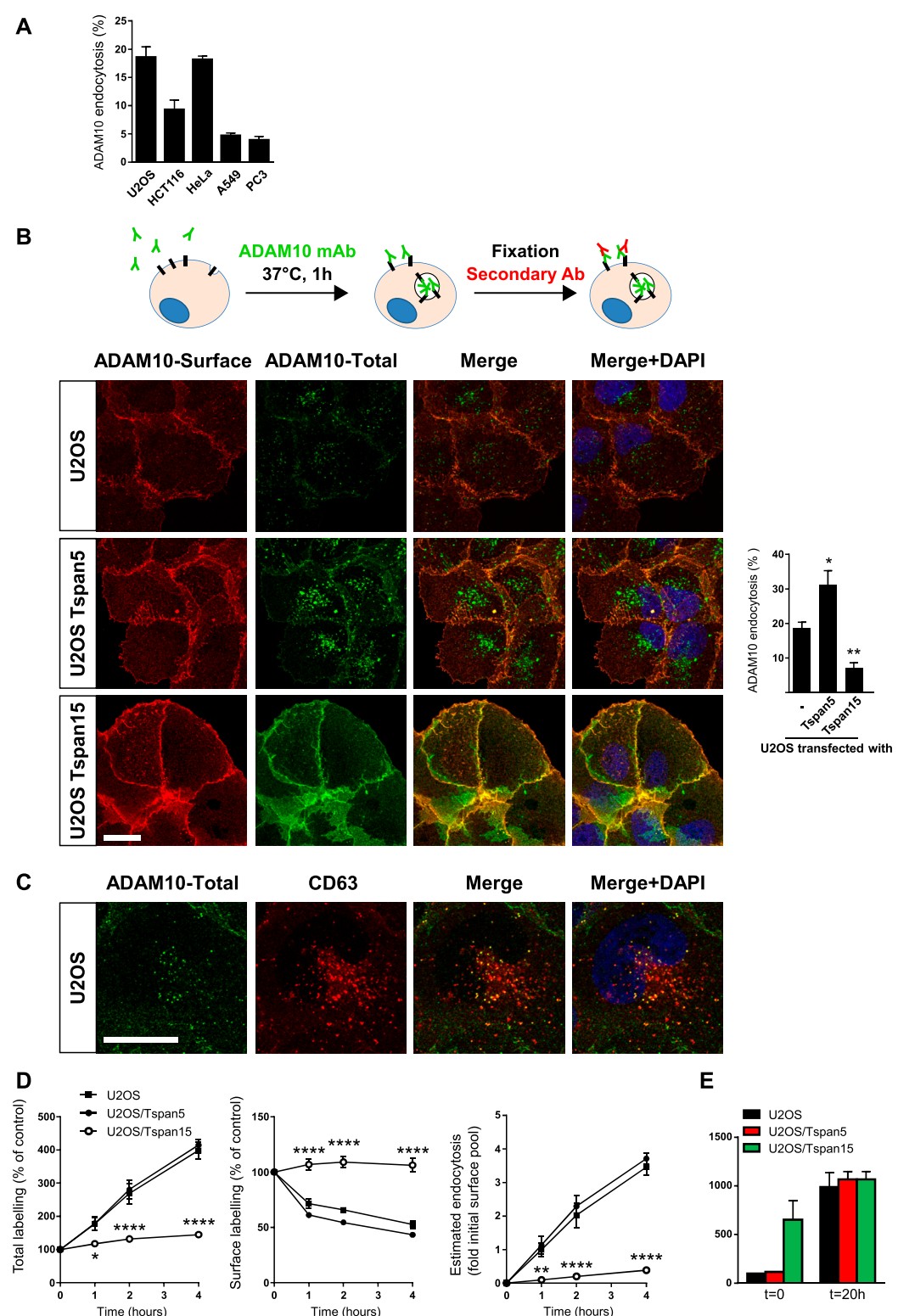

**Figure 2. Tspan5 and Tspan15 have an opposite effect on ADAM10 endocytosis in U2OS cells.**
**(A)** Quantification of ADAM10 endocytosis in various cell lines after incubation for 1 h with 10 μg/ml DyLight 650–labelled anti-ADAM10 mAb, using the confocal microscopy protocol. **(B)** U2OS cells transfected or not with Tspan5 or Tspan15 were incubated for 1 h with 10 μg/ml DyLight 650–labelled anti-ADAM10 mAb (green). After fixation, the surface pool of ADAM10 (red) was stained using an Alexa488-labelled secondary reagent. The intracellular pool is not labelled by the secondary reagent and, therefore, appears in green. A quantification is shown on the right. Data were statistically analyzed by a one-way ANOVA followed by a Dunnett's multiple comparisons test (*$P < 0.05$ and **$P < 0.01$ compared with WT, n = 5). Bar: 10 μm. **(C)** U2OS cells were incubated for 1 h with 10 μg/ml DyLight 650–labelled anti-ADAM10 mAb (green). After

indicating that in this time interval, a similar amount of ADAM10 had reached the membrane in all cells. Thus, the ADAM10 cell surface expression level at steady state is a poor reflection of the overall amount of ADAM10 having passed through the plasma membrane.

## In HeLa cells, Tspan5 and Tspan15 also differentially regulate ADAM10 endocytosis and half-life

To validate the previous findings in a different cell model, Tspan5 and Tspan15 were also expressed in HeLa cells. Both tetraspanins induced a similar twofold increase in ADAM10 surface expression levels (Fig 3A and Table 1). Again, in this model, Tspan15 inhibited ADAM10 endocytosis (Fig 3B). The absolute amount of ADAM10 endocytosed in Tspan5-transfected HeLa cells was higher than in parental cells, but the percentage of endocytosed ADAM10 was not significantly different. A large fraction of the endocytosed ADAM10 colocalized with Tspan5, suggesting that both proteins are endocytosed together (Fig 3C).

So far, the data show that Tspan15 expression is associated with both a higher stability and a diminished endocytosis of ADAM10. We reasoned that if the higher stability of the protein is a consequence of the stabilization at the cell membrane, the surface pool of ADAM10 should be more stable in the presence of Tspan15. To address this question, surface proteins were labelled with biotin, and the cells were incubated at 37°C for 15 h before lysis, or lysed directly to determine the initial level of surface protein. ADAM10 and CD81 were then immunoprecipitated, and the level of biotin-labelled protein was determined using streptavidin blotting (Fig 3D). As a control, the same membrane was probed with a CD81 or an anti-ADAM10 mAb. The level of biotin-labelled ADAM10 was reduced after 15 h by ~40–50% in HeLa cells and in HeLa/Tspan5 cells, but by less than 20% in HeLa/Tspan15 cells. In contrast, in both cell lines, the initial pool of CD81 was decreased by ~80% (Fig 3D). These data indicate that the surface pool of ADAM10 is more stable in the presence of Tspan15 than in the presence of Tspan5.

## Endogenous Tspan5 and Tspan15 differentially regulate ADAM10 endocytosis and stability

The preceding data showing that Tspan5 and Tspan15 differentially regulate ADAM10 trafficking and turn-over were obtained using transfected cells. To analyze the impact of endogenous Tspan5 and Tspan15, we generated Tspan5 and Tspan15 KO cells using the CRISPR/Cas9 gene editing technology.

In a first model, ablation of Tspan5 in U2OS cells did not modify ADAM10 surface levels, but reduced its internalization rate by 50%, further suggesting that Tspan5 stimulates ADAM10 endocytosis (Fig 4).

In PC3 cells, Tspan15 KO decreased ADAM10 surface level by 75%, whereas Tspan5 KO had little effect (Fig 5A). This was associated with a decrease in the total level of ADAM10, as shown by Western blot (Fig 5B) and with a slight increase in the intracellular pool of ADAM10 (Fig S3). This decrease in the total level of ADAM10 was not associated with a decrease in ADAM10 mRNA levels (data not shown) and was due at least in part to a shorter ADAM10 half-life (more than 8 h in PC3 cells versus less than 4 h in Tspan15 KO cells), as shown after cycloheximide treatment (Fig 5B). This was associated with an increased ADAM10 endocytosis rate, as shown by confocal microscopy and the time-dependent accumulation of the ADAM10 mAb inside the cells in Tspan15 KO cells, but not in parental cells (Fig 5C and D and Table 1). After 20-h incubation with the anti-ADAM10 mAb, the staining of Tspan15 KO PC3 cells was similar to that of parental and Tspan5 KO cells (Fig 5E), indicating that a similar amount of ADAM10 trafficked through the plasma membrane during this time period.

## Endogenous Tspan5 and Tspan15 compete for the interaction with ADAM10

Fig 5A shows that the ablation of Tspan15 in PC3 cells was associated with a higher Tspan5 surface expression level. To further investigate the relationship between ADAM10 and the two TspanC8 tetraspanins, we performed immunoprecipitations after cell lysis using either digitonin or Brij 97, two conditions that preserve the interaction of ADAM10 with TspanC8 tetraspanins (Fig 6). We used two different anti-Tspan5 mAbs: one recognizes Tspan5 whether it associates with ADAM10 or not (TS5-2), and the other one only recognizes the fraction of Tspan5 not associated with the protease (TS5-1r) (Saint-Pol et al, 2017a). In parental PC3 cells, after Brij 97 lysis, the anti-Tspan5 mAb TS5-2 immunoprecipitated and recognized by Western blot a 25–35-kD diffuse band, which was reinforced at ≈25 and 27 kD. The ≈25-kD band corresponds to a fraction of Tspan5 retained in the ER, not associated with ADAM10 as it was immunoprecipitated with TS5-1r and not with the anti-ADAM10 mAb (Saint-Pol et al, 2017a). A similar pattern was observed after digitonin lysis, except that there was no reinforcement at 25 kD. TS5-1r only immunoprecipitated the lower MW forms of Tspan5 (Fig 6).

The anti-Tspan5 mAbs TS5-2 immunoprecipitated only a small fraction of mature ADAM10 in parental cells. In contrast, most mature Tspan5 is associated with ADAM10, as shown by the comparison of the ADAM10 and TS5-2 IPs and by the lack of immunoprecipitation of the upper forms with TS5-1r (Fig 6).

There was more Tspan5 in the Tspan5 IP from Tspan15 KO cells (Fig 6), consistent with the higher surface expression level observed by flow-cytometry (Fig 5A). Despite a higher expression level, no Tspan5 could be immunoprecipitated with TS5-1r, which was

---

fixation, the cells were labelled in the presence of saponin with a CD63 mAb coupled to DyLight 550 (red). Bar: 10 μm. **(D)** U2OS cells transfected or not with Tspan5 or Tspan15 were incubated for the indicated time with 10 μg/ml DyLight 650–labelled anti-ADAM10 mAb. After detachment, the surface pool of ADAM10 was stained using a PE-labelled secondary reagent before flow cytometry analysis. The left and middle graphs show the evolution in time of DyLight 650 and PE fluorescence relative to cells stained at 4°C after detachment. The graph on the right shows an estimate of the amount of endocytosed ADAM10 in relation to the pool present at the cell surface in the corresponding untreated cells labelled at 4°C after detachment. Data were statistically analyzed by a one-way ANOVA followed by a Dunnett's multiple comparisons test (*P < 0.05; **P < 0.01; ****P < 0.0001 compared with WT, n = 4). Note that the error bars in this panel may not be visible for all samples at the scale used. **(E)** U2OS cells transfected or not with Tspan5 or Tspan15 were incubated or not for 20 h with 10 μg/ml DyLight 650–labelled anti-ADAM10 mAb. The graph shows the mean of the relative DyLight 650 fluorescence intensities in two experiments ± range (100 = intensity of untreated parental U2OS cells stained after detachment).

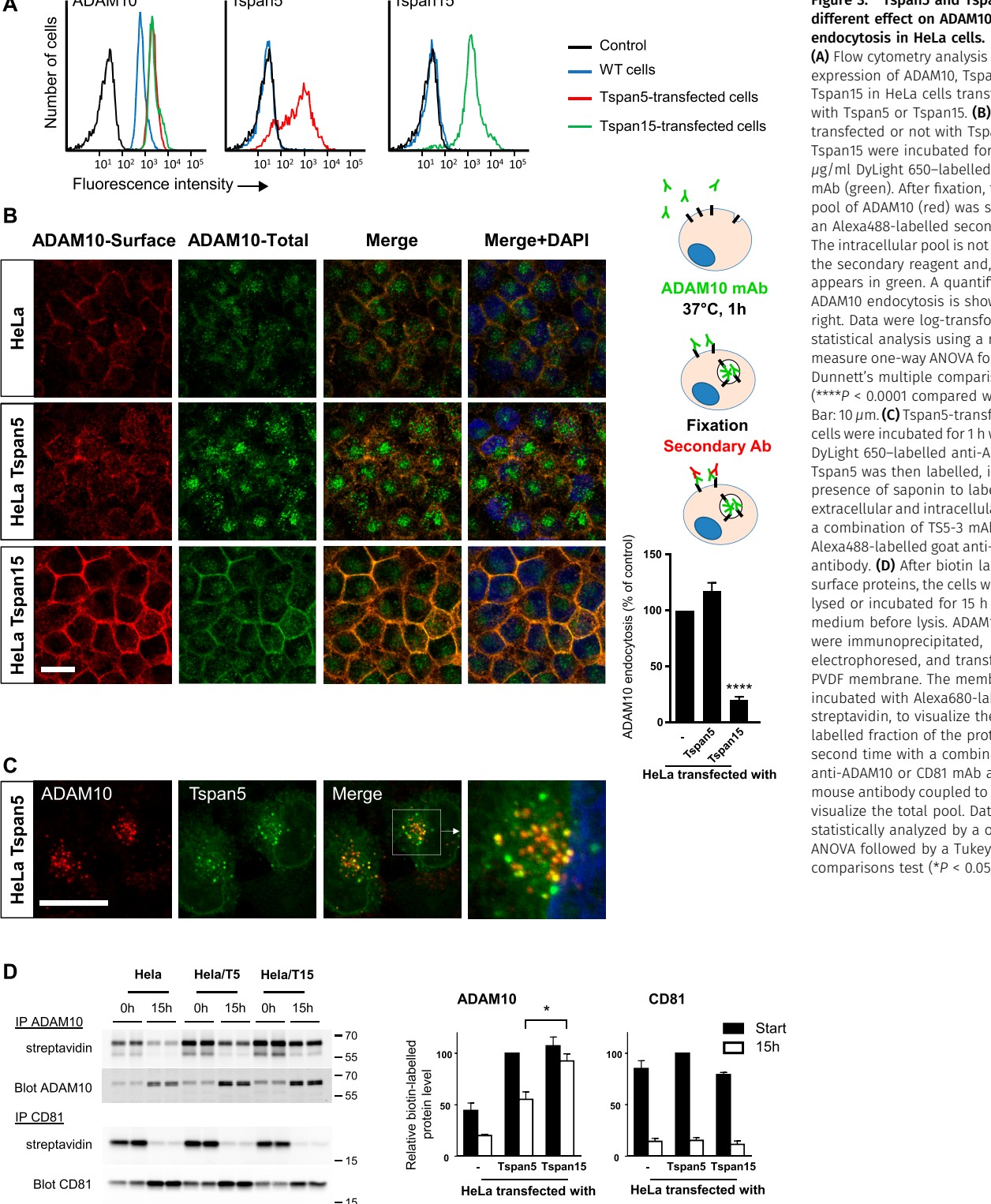

**Figure 3. Tspan5 and Tspan15 have a different effect on ADAM10 endocytosis in HeLa cells.**
**(A)** Flow cytometry analysis of the surface expression of ADAM10, Tspan5, and Tspan15 in HeLa cells transfected or not with Tspan5 or Tspan15. **(B)** HeLa cells transfected or not with Tspan5 or Tspan15 were incubated for 1 h with 10 µg/ml DyLight 650–labelled anti-ADAM10 mAb (green). After fixation, the surface pool of ADAM10 (red) was stained using an Alexa488-labelled secondary reagent. The intracellular pool is not labelled by the secondary reagent and, therefore, appears in green. A quantification of ADAM10 endocytosis is shown on the right. Data were log-transformed before statistical analysis using a repeated measure one-way ANOVA followed by a Dunnett's multiple comparisons test (****$P$ < 0.0001 compared with WT, n = 8). Bar: 10 µm. **(C)** Tspan5-transfected HeLa cells were incubated for 1 h with 10 µg/ml DyLight 650–labelled anti-ADAM10 mAb. Tspan5 was then labelled, in the presence of saponin to label both the extracellular and intracellular pools, using a combination of TS5-3 mAb and an Alexa488-labelled goat anti-mouse IgG2a antibody. **(D)** After biotin labelling of surface proteins, the cells were directly lysed or incubated for 15 h in complete medium before lysis. ADAM10 and CD81 were immunoprecipitated, electrophoresed, and transferred to a PVDF membrane. The membranes were incubated with Alexa680-labelled streptavidin, to visualize the biotin-labelled fraction of the proteins, and in a second time with a combination of anti-ADAM10 or CD81 mAb and anti-mouse antibody coupled to DyLight 800 to visualize the total pool. Data were statistically analyzed by a one-way ANOVA followed by a Tukey's multiple comparisons test (*$P$ < 0.05, n = 4).

consistent with the disappearance of the ≈25-kD band (after Brij 97 lysis) (Fig 6), and which indicates that most Tspan5 had left the ER and associated with ADAM10 in Tspan15 KO cells. Despite a reduced expression level of the mature form of ADAM10, Tspan5 was strongly

co-immunoprecipitated with ADAM10, and reciprocally TS5-2 co-immunoprecipitated a higher fraction of ADAM10 (Fig 6). Altogether, these data indicate that endogenous Tspan15 competes with Tspan5 for the association with ADAM10 and that the ablation of Tspan15

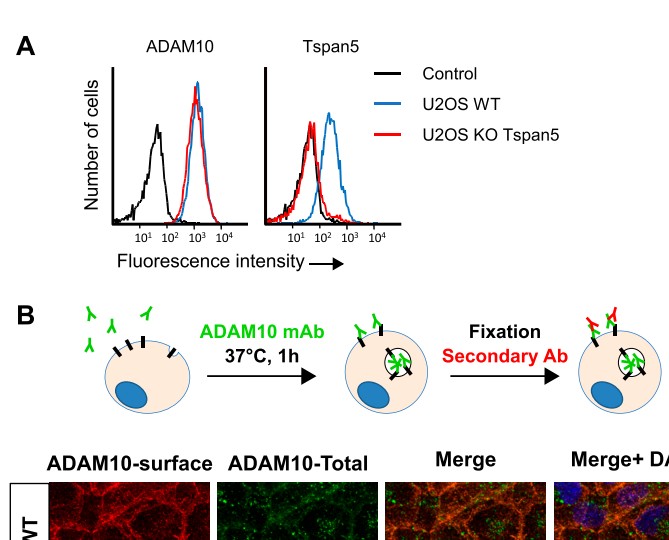

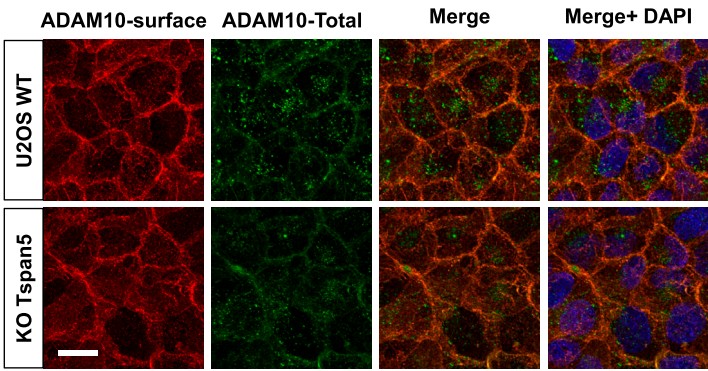

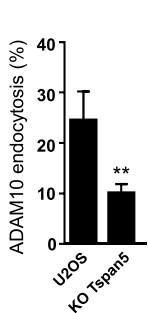

**Figure 4. Endogenous Tspan5 accelerates ADAM10 endocytosis in U2OS cells.**
**(A)** Flow cytometry analysis of the surface expression of ADAM10 and Tspan5 in WT U2OS cells or cells engineered to lack Tspan5 using the CrispR/Cas9 gene editing technology. **(B)** U2OS cells lacking or not Tspan5 were incubated for 1 h with 10 µg/ml DyLight 650–labelled anti-ADAM10 mAb. After fixation, the surface pool of ADAM10 (red) was stained using an Alexa488-labelled secondary reagent. The intracellular pool is not labelled by the secondary reagent and, therefore, appears in green. A quantification of ADAM10 endocytosis is shown on the right. Statistical analysis was performed using a ratio paired t test. (**P < 0.01 compared with WT, n = 3). Bar: 10 µm.

allows ADAM10 to associate with a higher fraction of Tspan5 and facilitate Tspan5 exit from the ER, leading to a higher Tspan5 expression level. In addition, there was more Tspan15 in the ADAM10 IP collected from Tspan5 KO cells than from parental cells (Fig 6), indicating that endogenous Tspan5 limits the association of Tspan15 with ADAM10.

## Analysis of the role of Tspan5 and Tspan15 palmitoylation

Tetraspanins are palmitoylated at several conserved juxtamembrane intracellular cysteines. Tspan15, in contrast to Tspan5, lacks such typical tetraspanin intracellular cysteines (Fig 7A). Because the differential effect of Tspan5 and Tspan15 on ADAM10 function and trafficking correlates with a different ability to associate with classical tetraspanins (Fig 1 and Jouannet et al (2016)) and palmitoylation of tetraspanins has been shown to contribute to their interaction with one another and to modulate some of their functions (Berditchevski et al, 2002; Charrin et al, 2002; Yang et al, 2002; Cherukuri et al, 2004), we analyzed whether Tspan5 and Tspan15 are differentially palmitoylated and whether this post-translational modification contributes to the regulation of ADAM10.

To test whether Tspan5 and Tspan15 are palmitoylated proteins, the cells were incubated with azido palmitic acid and palmitoylated proteins were labelled with biotin using click chemistry. As expected, Tspan5 efficiently incorporated palmitic acid and the Tspan5plm mutant, in which all four cysteines that are putative palmitoylation sites were changed to alanines, did not (Fig 7B), indicating that Tspan5 is exclusively palmitoylated at one or several of the mutated cysteines. Surprisingly, despite the absence of the

intracellular cysteines conserved in most tetraspanins, Tspan15 also efficiently incorporated palmitic acid (Fig 7B). A unique feature of Tspan15 is the presence of three cysteines at the end of its C-terminus (CCLCYPN). Mutation of the three cysteines to alanines in the CCLC motif strongly reduced the incorporation of palmitic acid, indicating that this motif constitutes a major palmitoylation site in Tspan15 (Fig 7B). These data suggest that the end of the C-terminus of Tspan15 is at least in a fraction of the molecules anchored to the membrane through palmitoylation of the CCLC motif. They also suggest that the interaction of tetraspanins with one another does not only depend on the overall palmitoylation level but may also require that this modification occurs at particular cysteines.

Both the Tspan5plm mutant and the Tspan15 CCLC mutant were expressed at the cell surface similarly to the WT proteins (data not shown). However, these mutations did not affect the influence of Tspan5 and Tspan15 on ADAM10 surface expression levels in U2OS cells or Notch activity (Fig 7C and D).

## Analysis of Tspan5 and Tspan15 domains responsible for their differential regulation of ADAM10 endocytosis and Notch signaling

To get further insights into the mechanisms whereby Tspan5 and Tspan15 differentially regulate ADAM10 endocytosis and Notch signaling, we generated a series of Tspan5/Tspan15 chimeras (Fig 8A) and expressed them in HeLa and U2OS cells. In both cell lines, the chimeras were expressed at a level similar to that of Tspan5 or Tspan15 (Fig S4). As shown in Fig 8B, a chimera in which the large extracellular domain (also referred to as the large extracellular

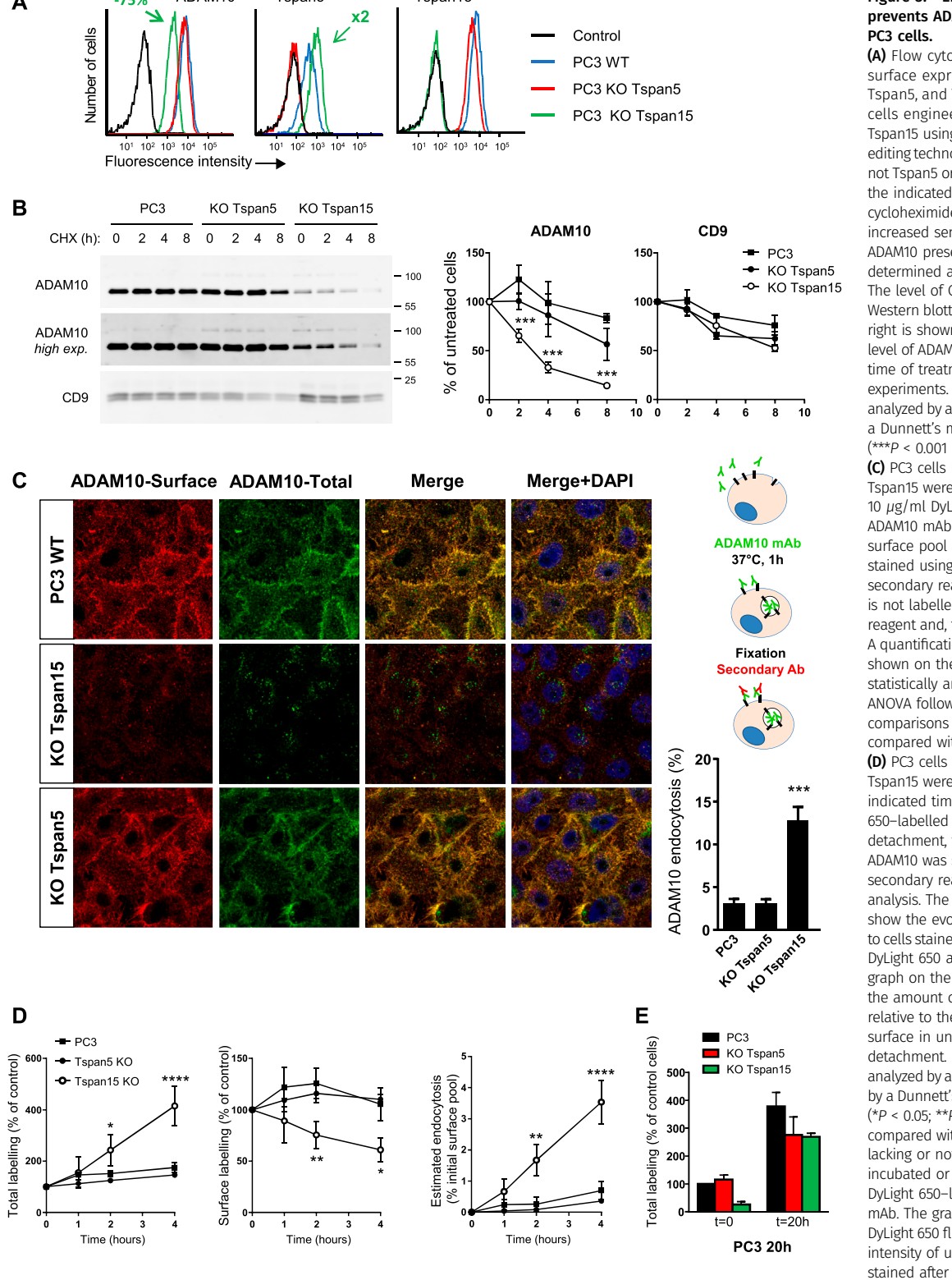

**Figure 5. Endogenous Tspan15 prevents ADAM10 endocytosis in PC3 cells.**
**(A)** Flow cytometry analysis of the surface expression of ADAM10, Tspan5, and Tspan15 in WT PC3 cells or cells engineered to lack Tspan5 or Tspan15 using the CrispR/Cas9 gene editing technology. **(B)** PC3 cells lacking or not Tspan5 or Tspan15 were incubated for the indicated time with 100 μg/ml cycloheximide (CHX) before lysis. For increased sensitivity, the amount of ADAM10 present in the lysates was determined after immunoprecipitation. The level of CD9 was determined by Western blotting on the lysates. On the right is shown a quantification of the level of ADAM10 and CD9 according to the time of treatment in three different experiments. Data were statistically analyzed by a one-way ANOVA followed by a Dunnett's multiple comparisons test (***$P$ < 0.001 compared with WT, n = 3). **(C)** PC3 cells lacking or not Tspan5 or Tspan15 were incubated for 1 h with 10 μg/ml DyLight 650–labelled anti-ADAM10 mAb (green). After fixation, the surface pool of ADAM10 (red) was stained using an Alexa488-labelled secondary reagent. The intracellular pool is not labelled by the secondary reagent and, therefore, appears in green. A quantification of ADAM10 endocytosis is shown on the right. Data were statistically analyzed by a one-way ANOVA followed by a Dunnett's multiple comparisons test (***$P$ < 0.001 compared with WT, n = 3). Bar: 10 μm. **(D)** PC3 cells lacking or not Tspan5 or Tspan15 were incubated for the indicated time with 10 μg/ml DyLight 650–labelled anti-ADAM10 mAb. After detachment, the surface pool of ADAM10 was stained using a PE-labelled secondary reagent before flow cytometry analysis. The left and middle graphs show the evolution in time, and relative to cells stained at 4°C after detachment, of DyLight 650 and PE fluorescence. The graph on the right shows an estimate of the amount of endocytosed ADAM10 relative to the pool present at the cell surface in untreated cells labelled after detachment. Data were statistically analyzed by a one-way ANOVA followed by a Dunnett's multiple comparisons test (*$P$ < 0.05; **$P$ < 0.01; ****$P$ < 0.0001 compared with WT, n = 3). **(E)** PC3 cells lacking or not Tspan5 or Tspan15 were incubated or not for 20 h with 10 μg/ml DyLight 650–labelled anti-ADAM10 mAb. The graph shows the relative DyLight 650 fluorescence intensities (100 = intensity of untreated parental cells stained after detachment; [n = 3]).

loop, LEL) of Tspan5 was replaced by that of Tspan15 (T5LEL15) stimulated ADAM10 endocytosis in HeLa cells, whereas the reverse chimera (T15LEL5) inhibited its endocytosis. Thus, the LEL of these tetraspanins, which probably mediate the interaction with ADAM10

(Noy et al, 2016; Saint-Pol et al, 2017a), do not contribute to their opposite effect on ADAM10 endocytosis. T15LEL5 conserved the ability of Tspan15 to inhibit Notch signaling, suggesting no specific role of the LEL in the regulation of Notch signaling (Fig 8C). The

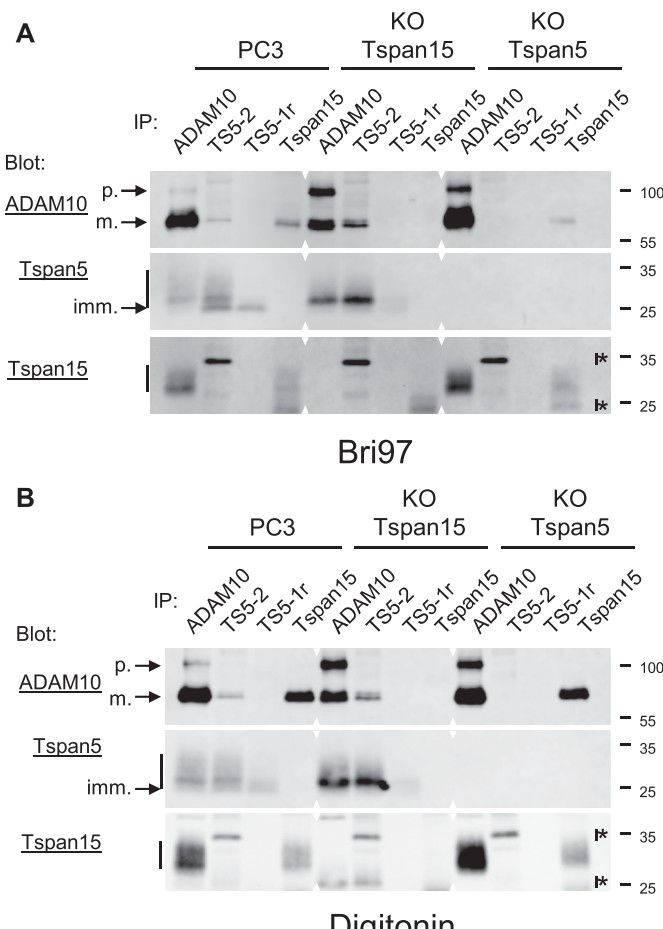

**Figure 6. Endogenous Tspan5 and Tspan15 compete for the association with ADAM10.**
**(A, B)** PC3 cells lacking or not Tspan5 or Tspan15 were lysed using Brij 97 (A) or digitonin (B) and immunoprecipitations were performed as indicated on the top of each lane. The composition of the immunoprecipitates was analyzed by Western blot using biotin-labelled anti-ADAM10 and Tspan5 mAb, or a non-labelled anti-Tspan15 mAb. Note that the bands revealed by the Tspan15 mAb in Tspan15 KO cells, and also present in the other cell types, are nonspecific bands that are also present when the membranes are incubated with the secondary reagent only (indicated by I*). The mature (m.) and proform (p.) forms of ADAM10, as well as the immature (imm.) form of Tspan5, are indicated by arrows.

reverse chimera (T5LEL15) also partially inhibited Notch signaling, but to a lower extent. However, the cells expressing this chimera had diminished ADAM10 surface levels (by ~20–30% according to the experiments) (Fig 8D), which is consistent with the finding that this chimera stimulates ADAM10 endocytosis more than Tspan5 in HeLa cells (Fig 8B). This is likely to be the cause of a lower Notch signaling and not because the fraction of ADAM10 associated with this chimera is not competent for Notch signaling. This decrease in ADAM10 surface level after transfection of this chimera was observed using two independently transfected populations, suggesting a specific effect of this chimera.

A major difference between Tspan5 and Tspan15 is the presence in Tspan15 of a cytoplasmic intracellular C-terminal domain unusually long for a tetraspanin (>30 amino acids, whereas tetraspanins including Tspan5 have generally less than 10–15 residues). Replacing the N- and C-cytoplasmic domains of Tspan5 by that of

Tspan15 (T5NC15) yielded a chimera that strongly inhibited ADAM10 endocytosis in HeLa cells (Fig 8B). Chimeras in which either the N- or the C-cytosolic domains were exchanged had intermediate effect on ADAM10 endocytosis. Altogether, these data indicate that the cytosolic domains of Tspan15 contribute to the ability of Tspan15 to inhibit ADAM10 endocytosis in HeLa cells. However, exchanging these domains by that of Tspan5 in Tspan15 (in T15C5, T15N5, and T15NC5) yielded chimeras that still inhibited ADAM10 endocytosis (Fig 8B), indicating that other parts of Tspan15 contribute to the inhibitory action of Tspan15.

When expressed in U2OS cells, all chimeras in which the cytoplasmic domains of Tspan5 and Tspan15 were exchanged stimulated an increase in ADAM10 surface expression level (about twofold) that was lower than that induced by Tspan15 (Tspan5 overexpression did not result in an increase in ADAM10 surface levels) (Fig 8D). At least in the case of T5C15 and T15C5, this was due to an internalization level intermediate between that observed in U2OS cells transfected with Tspan5 or Tspan15 (Fig 8E). These data further indicate that the cytosolic domains of Tspan5 and Tspan15 contribute to their opposite action on ADAM10 endocytosis and surface expression level, but that other parts of the molecules also play a role.

We also analyzed the effect of these chimeras on Notch signaling (Fig 8C). The replacement of either the N- or C-terminal cytosolic domains of Tspan15 by those of Tspan5 was sufficient to completely abolish the ability of Tspan15 to inhibit Notch signaling. In contrast, T5C15 had only a small inhibitory effect.

## ADAM10 stabilizes Tspan15 at the plasma membrane

The above data show that Tspan5 and Tspan15 have a positive and a negative impact, respectively, on ADAM10 endocytosis. Consistent with this finding, the Tspan5 mAb was efficiently internalized after 1 h at 37°C in HeLa/Tspan5 cells, and the internalized fraction colocalized with ADAM10, suggesting again that both proteins were internalized together (Fig 9A, top). However, two different behaviors were observed for Tspan15. The anti-Tspan15 mAb was not internalized in cells expressing moderate amounts of Tspan15 but was efficiently internalized in cells with the highest level of expression of Tspan15 (Fig 9A, middle and bottom). Importantly, this internalized fraction did not co-localize with ADAM10. We reasoned that in these latter cells, the level of Tspan15 would exceed that of ADAM10 and, therefore, hypothesized that only the fraction of Tspan15 not associated with ADAM10 was efficiently internalized. This would be consistent with the finding that the intracellular pool of Tspan15 in transfected U2OS cells does not co-localize with ADAM10 (Fig S1).

To validate the hypothesis that ADAM10 stabilizes Tspan15 at the cell surface, we generated ADAM10 KO PC3 cells. Although there was no change in Tspan15 mRNA levels (data not shown), there was a 60% decrease in Tspan15 surface expression levels (Fig 9B) in these cells. Moreover, whereas Tspan15 was poorly endocytosed after 1 h at 37°C in parental PC3 cells, it was efficiently endocytosed in ADAM10 KO cells (Fig 9C). We conclude that ADAM10 stabilizes Tspan15 at the cell surface.

This finding may explain why transfection of Tspan5 in PC3 cells led to a ~40% decrease of Tspan15 surface expression (Fig S2B). The overexpressed Tspan5 probably competes to some extent with Tspan15 for the interaction of ADAM10, yielding a fraction of Tspan15

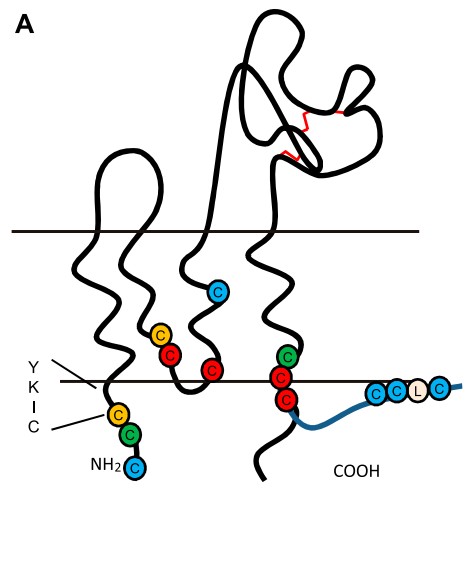

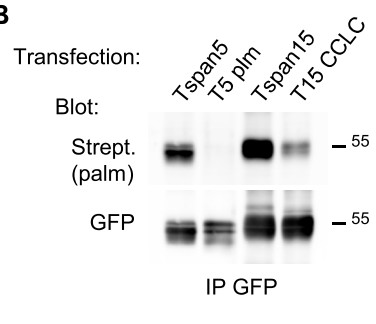

**A**

**B**

Transfection:

Blot:

Strept.
(palm)

GFP

— 55

— 55

IP GFP

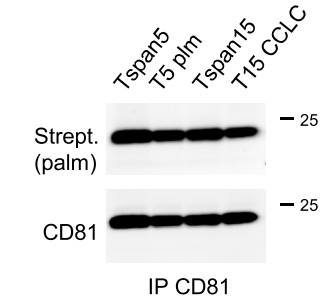

Strept.
(palm)

CD81

— 25

— 25

IP CD81

**Figure 7. Analysis of Tspan5 and Tspan15 palmitoylation.**
**(A)** Schematic structure of tetraspanins, with indication of intracellular or transmembrane cysteines. Cysteines in yellow, red and green correspond to those present in Tspan5 and other tetraspanins, in other tetraspanins but not Tspan5 or Tspan15, and in Tspan5 but no in other tetraspanins, respectively. Cysteines in blue are those from Tspan15. Note that none of them aligns with cysteines present in classical tetraspanins. **(B)** GFP-tagged versions of Tspan5, Tspan5plm, Tspan15, and the Tspan15 CCLC mutant in which the three cysteines of the CCLC motif were mutated to alanines were expressed in U2OS cells. Their ability to incorporate palmitate moieties was determined using click chemistry as described in the Materials and Methods section. The palmitates attached to the proteins were visualized using Alexa680-labelled streptavidin (Strept.). As a control, the level of palmitoylation of endogenous CD81 was also determined. **(C)** Relative ADAM10 expression in U2OS cells expressing untagged Tspan5, Tspan5plm, Tspan15, or Tspan15 CCLC was determined by flow cytometry (n = 3). **(D)** U2OS-N1 cells stably expressing or not untagged Tspan5, Tspan5plm, Tspan15, or Tspan15 CCLC were analyzed for Notch activity after co-culture with OP9 cells expressing the Notch ligand DLL-1 (n = 3). Data were log-transformed before statistical analysis using a repeated measures one-way ANOVA. Each transfectant was compared with parental cells using the Dunnett's multiple comparisons test. $***P < 0.001$; $****P < 0.0001$.

**C**

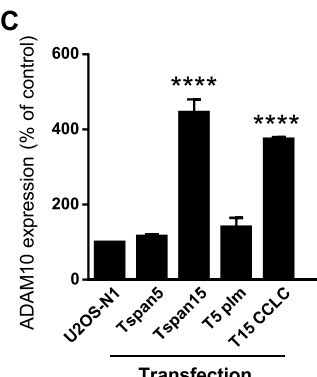

**D**

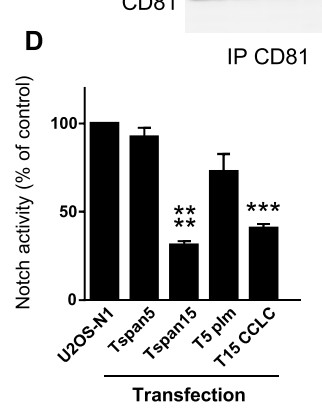

not associated with ADAM10 that undergoes rapid internalization and degradation.

## Discussion

The trafficking and function of the metalloprotease ADAM10 is regulated by the members of the TspanC8 subgroup of tetraspanins. Whereas all TspanC8 members allow ADAM10 to exit from the ER, Tspan10 and Tspan17 were shown to target ADAM10 to late endosomes, and the remaining four members target ADAM10 to the plasma membrane (Dornier et al, 2012; Haining et al, 2012; Prox et al, 2012). Here, we show using different models that Tspan5 and Tspan15 differentially regulate ADAM10 endocytosis and turnover and highlight a complex cross-regulation of ADAM10, Tspan5, and Tspan15 expression levels.

### Tspan5 and Tspan15 differentially regulate ADAM10 surface expression levels and turnover

Although both Tspan5 and Tspan15 target ADAM10 to the plasma membrane, Tspan15 is associated with higher ADAM10 surface expression levels: indeed, transfection of Tspan15 in a Tspan5/Tspan14–positive cell line (U2OS) induced a large increase in ADAM10 surface expression levels, whereas transfection of Tspan5 in a Tspan5/Tspan15–positive cell line (PC3) decreased the ADAM10 surface level. In addition, we observed that the surface staining of PC3 cells with a Tspan15 mAb was nearly ten times higher than that observed with a Tspan5 mAb, suggesting a much higher expression level of Tspan15 at the protein level. Accordingly, Tspan5 knockout had little effect on ADAM10 expression, whereas Tspan15 knockout reduced by 75% the surface expression of ADAM10 suggesting that most surface ADAM10 molecules are associated with Tspan15 at the steady state level. In both the U2OS and PC3 models, we have shown that the higher ADAM10 surface levels observed in the presence of Tspan15 reflects a higher total expression level, which is explained at least in part by a longer ADAM10 half-life.

Contrasting with U2OS and PC3 cells, Tspan5 transfection in HeLa cells yields a twofold increase in the ADAM10 surface level. This is most likely because in this cell line, ADAM10 is retained to a large extent in the ER because of a limiting amount of TspanC8 (Dornier et al, 2012). Thus, the effect of stimulating ADAM10 exit from the ER (as previously shown for GFP-tagged Tspan5 and the other TspanC8 tetraspanins [Dornier et al, 2012]) may ultimately yield to an

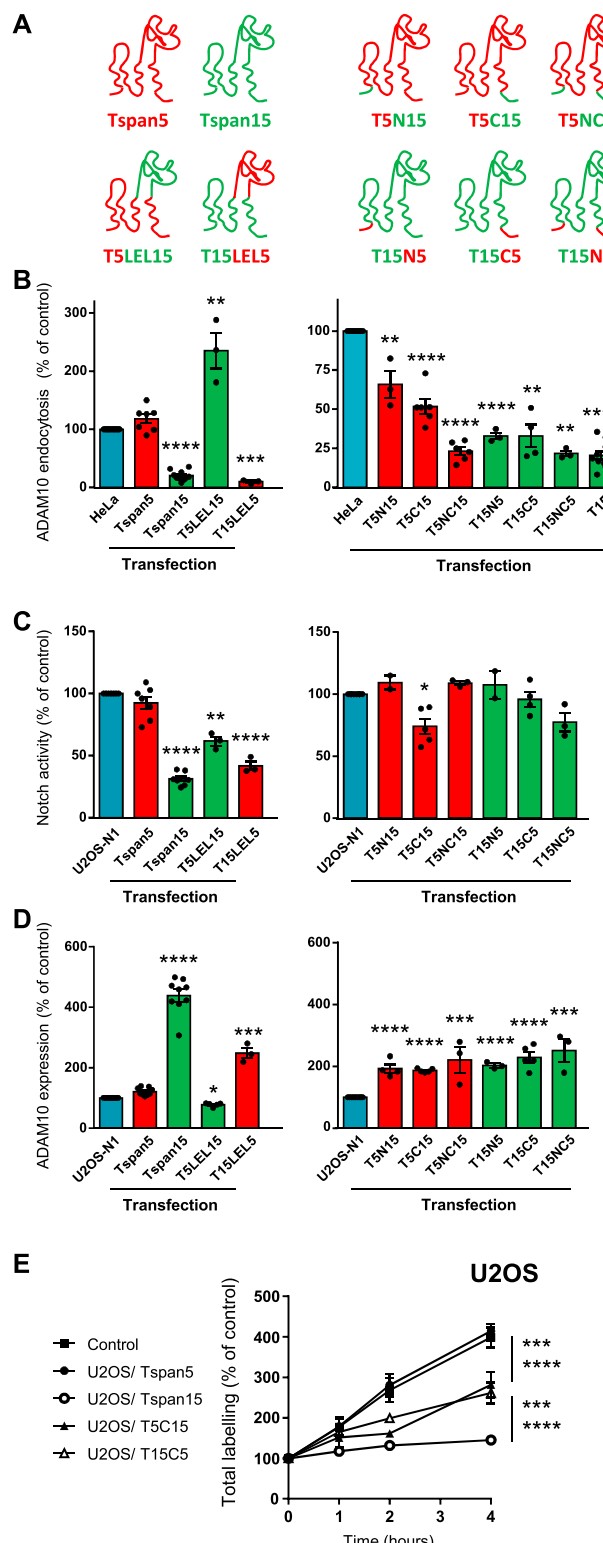

increased ADAM10 level, despite a negative impact of Tspan5 on ADAM10 surface levels. In addition, transfection of Tspan15 did not induce an ADAM10 expression level higher than that induced by Tspan5. This is likely to be due to a limiting amount of Tspan15. Indeed, in this cell line, the expression levels of transfected Tspan5 and Tspan15 were lower than that obtained in U2OS cells and similar to that of ADAM10 (Fig 3A). In favor of this hypothesis, after cell sorting, the few cells expressing the highest levels of Tspan15 showed higher ADAM10 surface levels (data not shown).

## Tspan5 and Tspan15 differentially regulate ADAM10 endocytosis and residence time at the plasma membrane

Our data indicate that the longer half-life of ADAM10$_{Tspan15}$ is at least in part the consequence of a longer residence time at the cell surface and that the analysis of ADAM10 cell surface expression at the steady state does not reflect the overall amount of ADAM10 having passed through the plasma membrane. Indeed, we have shown in several cellular models that ADAM10$_{Tspan5}$ undergoes faster endocytosis than ADAM10$_{Tspan15}$, and that after endocytosis, ADAM10 is targeted to CD63-positive compartments which include late endosomes and lysosomes. Notably, incubation of the cells with an anti-ADAM10 mAb showed a time-dependent accumulation of the mAbs inside the cells, which was inhibited by the expression of Tspan15 (either in U2OS-transfected cells or WT PC3 cells). After 20-h incubation with the anti-ADAM10 mAb at 37°C, staining of Tspan15 KO PC3 cells is comparable with that of parental cells or Tspan5 KO cells, indicating that although the ADAM10 level is strongly reduced in Tspan15 KO PC3 cells, the amount of ADAM10 trafficking through the plasma membrane is similar in all cell types. Thus, the fact that ADAM10 is mainly associated with Tspan15 at a steady state in PC3 cells is not due to a lower Tspan5 level but mainly if not exclusively to the shorter residence time of ADAM10$_{Tspan5}$ at the plasma membrane.

Tspan15 was shown to stabilize ADAM10 and to inhibit its endocytosis in all cell lines tested and using all assays performed. This, together with the finding that its deletion is associated with a decrease of the mature form of ADAM10 in the brain (Seipold et al, 2018), suggests a general function of Tspan15 for the stabilization of ADAM10 at the cell surface. In contrast, the stimulatory effect of

**Figure 8.  Analysis of Tspan5/Tspan15 chimeras.**
**(A)** Schematic representation of the various chimera used. **(B)** Analysis using the confocal microscopy protocol of ADAM10 endocytosis in HeLa cells transfected with Tspan5, Tspan15, or the different chimeras. Data are expressed as a function of the endocytosis rate measured in parental HeLa cells. **(C)** Analysis of Notch signaling in U2OS cells transfected with the different chimeras and

stimulated with OP9 cells expressing the Notch ligand DLL1. The data are expressed as a percentage of the signal observed for control U2OS cells. **(D)** Relative ADAM10 expression in U2OS cells expressing the various chimeras, determined by flow-cytometry. **(E)** U2OS cells transfected or not with Tspan5 or Tspan15, or the chimeras T5C15 and T15C5 were incubated for the indicated time with 10 µg/ml DyLight 650–labelled anti-ADAM10 mAb. After detachment, the surface pool of ADAM10 was stained using a PE-labelled secondary reagent before flow-cytometry analysis. The graph shows the evolution in time, and relative to cells stained at 4°C after detachment, of DyLight 650 fluorescence. This figure summarizes the data obtained in different experiments in which HeLa cells and Tspan5- and Tspan15-transfected cells were always analyzed as a reference. In (E), data were statistically analyzed after normalization using a one-way ANOVA followed by a Dunnett's multiple comparisons test. In the other panels, data were log-transformed before statistical analysis on the various sets of experiments using a repeated measures one-way ANOVA. Each transfectant was compared with parental cells using the Dunnett's multiple comparisons test. *$P < 0.05$, **$P < 0.01$, ***$P < 0.001$, ****$P < 0.0001$.

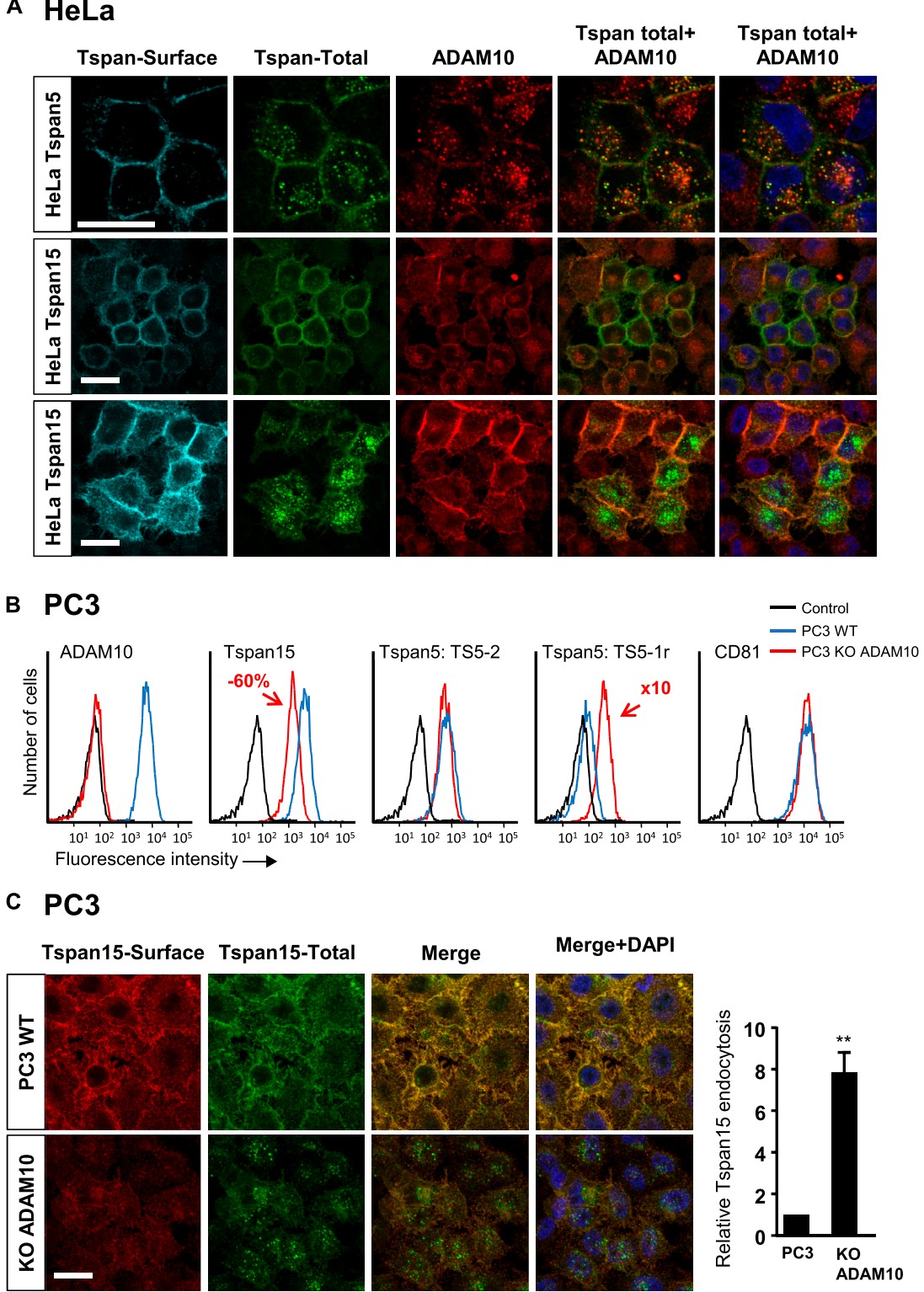

**Figure 9. ADAM10 regulates Tspan15 endocytosis.**
**(A)** HeLa cells transfected or not with Tspan5 or Tspan15 were incubated for 1 h with 10 μg/ml anti-Tspan5 (TS5-2) or Tspan15 (5F4) mAbs. After fixation, the surface pool of these tetraspanins (cyan) was stained using an Alexa568-labelled anti-subclass antibody. The cells were then incubated with the anti-ADAM10 mAb in the presence of saponin to label both the extracellular and intracellular pools. In a final step, the cells were incubated with appropriate anti-subclass antibodies to visualize ADAM10 (red) and tetraspanins (green). Note that the intracellular pool of tetraspanins visualized in this experiment corresponds to the fraction internalized during the time of the experiment, whereas that of ADAM10 corresponds to the total ADAM10 fraction. Bar: 10 μm. **(B)** Flow cytometry analysis of the surface expression of ADAM10, Tspan5,

Tspan5 on ADAM10 endocytosis in transfected or knockout cells was observed using the confocal microscopy approach, but was not confirmed using the flow-cytometry approach, and was not associated with a shorter half-life. The reasons for these discrepancies are unknown. Nevertheless, the analysis of Tspan5/Tspan15 chimeras in the different assays performed shows that ADAM10 can undergo a rate of endocytosis intermediate between that of ADAM10$_{Tspan5}$ and ADAM10$_{Tspan15}$. Because Tspan14 is the major TspanC8 tetraspanin expressed by HeLa and U2OS cells (besides Tspan5 in this latter case), it is possible that ADAM10$_{Tspan14}$ is endocytosed at a rate intermediate between that of ADAM10$_{Tspan5}$ and ADAM10$_{Tspan15}$. Further work will be required to test this hypothesis when mAbs to Tspan14 are available.

The endocytosis assays were performed using an antibody uptake assay, raising the question of the influence of the mAbs on ADAM10 endocytosis. The finding that incubation at 37°C with the anti-ADAM10 mAbs led to a decrease in ADAM10 at the surface of cells lacking Tspan15, that is, in U2OS and U2OS/Tspan5 cells and in tspan15 KO PC3 cells (by about 40–50% after 4 h, a similar decrease was observed after 24-h incubation), indicates that the mAb accelerates to some extent the rate of ADAM10 endocytosis. Nevertheless, several lines of evidence indicate that the internalization of the mAb mirrors the endocytosis of ADAM10 under normal conditions. First, the decrease at the cell surface of U2OS cells is for the most part observed when comparing the labelling after cell detachment with that observed after 1 h at 37°C, suggesting that cell detachment may accelerate endocytosis of antibody-bound ADAM10 in these cells. This could explain why we observe after 1-h incubation ~20% endocytosis by confocal microscopy and >50% using the flow cytometry protocol. Second, at time points later than 1 h, both in U2OS cells and Tspan15 KO PC3 cells, the mAbs continue to undergo effective internalization, accumulating into the cells at a much higher level than the initial surface level, whereas the surface level decreases proportionally much less.

The finding that overexpression of Tspan5 in U2OS cells does not induce an increase in the ADAM10 surface level contrasts with our previous study using GFP-tagged Tspan5 and Tspan15, which induce a three and five times increase in ADAM10 surface levels, respectively (Jouannet et al, 2016). This is most likely to be due to the stabilization of the Tspan5/ADAM10 complex at the cell surface by the GFP moieties, as expression of GFP-tagged Tspan5 in U2OS cells partially inhibited ADAM10 endocytosis (although not as much as Tspan15, data not shown). This highlights the importance of working with untagged proteins whenever possible.

### Potential mechanisms whereby Tspan5 and Tspan15 regulate ADAM10 endocytosis

We have started to study how Tspan5 and Tspan15 differentially affect ADAM10 surface levels by analyzing the activity of Tspan5/15 chimeras. Our data indicate a role for the cytosolic domains of Tspan5 and Tspan15 but also indicate that other domains contribute to their effect on ADAM10 endocytosis. This complex situation contrasts with the recent finding that replacing the C-terminus of Tspan15 by that of Tspan33 was sufficient to target Tspan15 and ADAM10 to apical junctions (Shah et al, 2018).

It has been suggested that the intracellular domain of ADAM10 interacts with the clathrin adaptor AP2 through a non-canonical AP2-binding motif that was shown to regulate to some extent the internalization of an IL2Rα/ADAM10 chimera (Marcello et al, 2013). Because the C-ter of Tspan15 is longer than that of Tspan5, one may hypothesize that this domain prevents the interaction of ADAM10 with one or several components of the endocytic machinery, and as a consequence, its internalization. However, the finding that T15C5 and T15NC5 remain inhibitors of ADAM10 endocytosis argues against this hypothesis. This result also suggests that the Tspan5 C-terminus does not contain a strong endocytic motif, which is coherent with the absence of tyrosine-based or other known endocytic motifs in this domain.

The longer residence time of ADAM10$_{Tspan15}$ at the plasma membrane could be due to a different recycling rate. In this regard, it was shown that Rab14 and its Guanine nucleotide exchange factor FAM116A play a role in recycling membrane proteins, and that depletion of these molecules in A549 cells, which strongly express Tspan15 (data not shown), induced the accumulation of ADAM10 in an intracellular transferrin-positive compartment (Linford et al, 2012). Alternatively, Tspan15 may dock ADAM10 at the plasma membrane. More exactly, because in the absence of ADAM10, but not in its presence, Tspan15 is efficiently endocytosed, the ADAM10/Tspan15 complex may be retained at the plasma membrane by interacting with discrete membrane proteins. In this regard, Tspan33 was shown to target ADAM10 to apical junctions in polarized epithelial cells through the interaction of its C-terminal domain with the adherens junction protein PLEKHA7 (Shah et al, 2018). In the same study, Tspan15 was shown to target ADAM10 to lateral contacts of polarized cells. Although we did not use polarized cells in this study, we observed that after incubation with the anti-ADAM10 mAb at 37°C, ADAM10 decorated lateral contacts in HeLa/Tspan15 more regularly than in HeLa/Tspan5 or parental HeLa cells. Thus, the ADAM10/Tspan15 complex may be stabilized at the plasma membrane through proteins that target the complex to lateral contacts. It should, however, be noted that extensive cell–cell contacts are not required for Tspan15 to inhibit ADAM10 endocytosis because this was also observed in dispersed cells in which cell–cell contacts are minimal (Fig 2 and data not shown).

### Cross-regulation of ADAM10, Tspan5, and Tspan15 expression levels

By demonstrating that Tspan5 and Tspan15 regulate ADAM10 endocytosis, this study reveals another level of ADAM10 regulation by

Tspan15, and CD81 in WT PC3 cells or cells engineered to lack ADAM10 using the CrispR/Cas9 gene editing technology. The mAb TS5-2 recognizes all Tspan5 molecules, whereas the mAb TS5-1r recognizes only the fraction of Tspan5 not associated with ADAM10. **(C)** PC3 cells lacking or not ADAM10 were incubated for 1 h with 10 μg/ml anti-Tspan15 mAb. The surface pool of Tspan15 (red) was stained using an Alexa488-labelled secondary reagent before fixation. In a second step, the endocytosed fraction (green) is labelled together with the surface pool in the presence of saponin using an Alexa647-labelled secondary antibody. A quantification of ADAM10 endocytosis is shown on the right. Data were statistically analyzed by a ratio paired *t* test. **P < 0.01 (n = 3). Bar: 10 μm.

TspanC8 tetraspanins. This study also extends the demonstration that ADAM10, in turn, regulates the trafficking of TspanC8. We have indeed previously demonstrated that ADAM10 facilitates Tspan5 exit from the ER, although it is not absolutely required (Saint-Pol et al, 2017a). This likely explains why a fraction of ectopically expressed Tspan5 is retained in the ER (see Fig S1 and Saint-Pol et al (2017a)). In contrast, we never observed an ER-like labelling of Tspan15 after overexpression (as judged by the absence of thin perinuclear labelling, Fig S1; [Dornier et al, 2012]), suggesting that ADAM10 might not regulate Tspan15 expression at this level. Rather, we demonstrate that the expression of Tspan15 is regulated at the plasma membrane because Tspan15 endocytosis is much faster in the absence of ADAM10, leading to a decrease in Tspan15 expression levels. Thus, it is probably the ADAM10/Tspan15 complex that is retained at the plasma membrane and not Tspan15 alone that would in turn retain ADAM10. Further work will be necessary to determine whether it is the ADAM10/Tspan5 complex that undergoes fast endocytosis.

We have also shown that overexpression of either Tspan5 or Tspan15 leads to a decrease in the expression level of the other tetraspanin. Conversely, ablation of Tspan15 in PC3 cells leads to a higher Tspan5 expression level. The simplest explanation for these results is that the competition between these tetraspanins for the association with ADAM10 may yield a pool of free tetraspanins which is retained in the ER in the case of Tspan5 or is quickly endocytosed and eliminated in the case of Tspan15. This hypothesis is supported by the observation that overexpression of Tspan15 in U2OS cells strongly diminishes the association of Tspan5 with ADAM10, and the finding that ablation of Tspan15 in PC3 cells promotes the association of Tspan5 with ADAM10 and its exit from the ER.

### Implications for ADAM10 function

The LEL of tetraspanins is one of the most variable domains and makes an important contribution to specific tetraspanin functions, notably by interacting with different partner proteins (Charrin et al, 2009, 2014). In this regard, the LEL of Tspan14 was shown to contribute to the interaction with ADAM10, and mutations of TspanC8-specific motifs in the LEL of Tspan5 and Tspan15 were shown to prevent the interaction with ADAM10 and to cause retention of the mutant in the ER (Noy et al, 2016; Saint-Pol et al, 2017a). In addition, truncation mutants and ADAM10/17 chimeras have shown that ADAM10 interacts with TspanC8 through its ectodomain and indicated that different TspanC8 engage ADAM10 in subtly different ways. Tspan15 notably associated better with a deletion mutant of ADAM10 comprising only the stalk (membrane proximal) region which led to the suggestion that it could regulate ADAM10 substrate selectivity by constraining it into a different conformation (Noy et al, 2016). In this study, using chimeras in which the LEL were swapped, we have shown that the LEL of Tspan15 does not support its ability to inhibit Notch signaling, excluding a mechanism involving a change in conformation. However, we observed that the chimera in which we replaced the LEL of Tspan5 by that of Tspan15 stimulated ADAM10 endocytosis better than Tspan5 in HeLa cells and decreased ADAM10 surface levels in U2OS cells. A possible explanation for this finding is that the Tspan15 LEL on a Tspan5 backbone yields

a chimera having a better affinity for ADAM10 than Tspan5, resulting in higher endocytosis.

It was recently demonstrated that inhibition of dynamin-mediated endocytosis of Notch1 reduced the colocalization of Notch1 with ADAM10 and the formation of the ADAM10-dependent cleavage product of Notch1, suggesting that ADAM10 may process Notch in an intracellular compartment (Chastagner et al, 2017). The discovery that Tspan15 was a negative regulator of both Notch signaling and ADAM10 endocytosis was consistent with this hypothesis. However, the finding that Tspan5/Tspan15 chimeras that inhibit ADAM10 endocytosis to some extent are not inhibitors of Notch signaling suggests that strong Tspan5-mediated ADAM10 endocytosis is not a requirement for ADAM10-dependent Notch signaling.

Remarkably, the replacement of either the N or the C cytoplasmic tails of Tspan15 by that of Tspan5 (in T15C5 and T15N5) yielded chimeras which had an intermediate effect on ADAM10 surface levels in U2OS cells and were not inhibitors of Notch signaling. Based on these data and the previous demonstration that Tspan5 and Tspan15 differentially regulate ADAM10 membrane compartmentalization (Jouannet et al, 2016), we suggest that the cytosolic domains of Tspan15 contribute to the stabilization of ADAM10 at cell surface in a membrane environment not permissive for the cleavage of Notch. Tspan15 may not directly prevent Notch from being recognized or cleaved by ADAM10, but rather may separate ADAM10 in a different membrane environment through the interaction with intracellular molecules.

### Concluding remarks

In conclusion, this study demonstrates that Tspan5 and Tspan15 differentially affect ADAM10 turn-over by regulating ADAM10 endocytosis. The relative expression levels of these tetraspanins has a major impact on the expression level of ADAM10 at a steady state so that this measure is a poor reflection of the total amount of the protease produced in different cell types. We have also shown that their cytosolic domains are important for their differential effect on ADAM10 endocytosis but that other parts of the molecules play a role. The study also shows that in turn ADAM10 regulates Tspan15 endocytosis, pointing out the necessity to work with endogenous proteins whenever possible. Further work will be necessary to determine why different pools of ADAM10 with different turnovers coexist in the cells. We suggest that the long residence time of ADAM10$_{Tspan15}$ at the plasma membrane is a consequence of the interaction with yet unknown proteins that stabilize this complex at lateral contacts, where it could process certain cadherins and perhaps other substrates. Concerning Tspan5, the short membrane residence time of ADAM10$_{Tspan5}$ may prevent ADAM10 from processing discrete substrates until appropriate, for example, after it is stabilized upon proper signaling. This may also allow ADAM10 levels to be increased in a transient manner in response to a particular signal. This may be especially true in the nervous system where Tspan5 is abundantly expressed (Garcia-Frigola et al, 2000) and ADAM10 plays a role in synaptic plasticity (Saftig & Lichtenthaler, 2015). In another hypothesis, Tspan5 may function to ensure that a certain amount of ADAM10 is present in the endocytic system, allowing cleavage of discrete ADAM10$_{Tspan5}$ substrates in this

compartment. The differential effect of Tspan5 and Tspan15 on ADAM10 endocytosis may be used to lower the expression levels and thereby the activity of particular pools of ADAM10.

# Materials and Methods

### Antibodies, plasmids, and mutagenesis

The mAb directed to human ADAM10 (11G2), CD9 (TS9), CD81 (TS81), CD63 (TS63), and Tspan5 (TS5-2, TS5-3, and TS5-1r) were generated in our laboratory (Charrin et al, 2001; Saint-Pol et al, 2017a; Arduise et al, 2008). The mAbs to Tspan15 (5F4 (IgG2b) and 5D4 (IgG1)) are described in a separate publication (Koo et al, 2019 Preprint). Labelling of antibodies with DyLight 650 or 550 N-Hydroxysuccinimide Esters was performed according to the manufacturer's instructions (Thermo Fisher Scientific). All fluorochrome-labelled anti-Ig antibodies were from Thermo Fisher Scientific. The plasmids encoding Tspan5 and Tspan15 fused to GFP were previously described (Dornier et al, 2012). After PCR amplification, the inserts were subcloned into the pCDNA3/hygro vector (Thermo Fisher Scientific). The various Tspan5/Tspan15 chimeras were generated using standard PCR-based approaches and subcloned in pCDNA3/hygro. The swaps in the chimeras were made at residues conserved in Tspan5 and Tspan15. T15LEL5 in which the LEL of Tspan15 is replaced by that of Tspan5 and the reciprocal construct T5LEL15 have been previously described (Saint-Pol et al, 2017a). The sequences of the swap sites at the C-terminus are Tspan15-LLPQFL**G**ICLAQN-Tspan5 for T15C5 and Tspan5-ALLQIF**G**VLLTLL-Tspan15 for T5C15. The sequences of the swap sites at the N-terminus were Tspan15-FSYLWL**K**YFIFGF-Tspan5 for T5N15 and T5NC15 and Tspan5-PEVSCCI**K**FSLII-Tspan15 for T15N5 and T15NC5 (the conserved amino acid where the swap was made is shown in boldface and is underlined).

### Quantitative RT-PCR

Total RNA fraction was isolated from $10^6$ cells using TRIzol (Thermo Fisher Scientific) according to the manufacturer's instructions. cDNA synthesis was performed from 1 $\mu$g of total RNA using 200u of SuperScript III Reverse Transcriptase (Invitrogen) primed with random hexamer (Promega), in a 20-$\mu$l reaction volume. Quantitative real-time PCRs (qPCRs) were then carried out on duplicates in a final volume of 25 $\mu$l containing 2× GoTaq qPCR Master Mix (SYBR Green) from Promega, 0.4 $\mu$M each forward and reverse primers and 1 $\mu$l cDNA. Quantification was performed with the Mx3005P QPCR Systems and MxPro software (Agilent). The values were normalized to rpl38 expression according to the ΔCt method. The oligonucleotides used for amplification have been previously described (Dornier et al, 2012).

### Cell culture and generation of cells expressing or lacking tetraspanins and ADAM10

OP9 cells expressing or not the human Notch ligand DLL-1 (OP9-DLL-1) (Six et al, 2004) were cultured in αMEM supplemented with 10% FCS and antibiotics. The human osteosarcoma cell line U2OS expressing human Notch1 (referred to as U2OS in this article) (Moretti et al, 2010) and prostate carcinoma cell line PC3 and the cervical carcinoma cell line HeLa were cultured in DMEM supplemented with 10% FCS and antibiotics. These cell lines have been previously described in terms of TspanC8 mRNA expression levels (Dornier et al, 2012; Jouannet et al, 2016). Transfection was performed using Fugene HD according to the manufacturer's instruction. Cells stably expressing the various constructs were isolated after selection with 100 $\mu$g/ml hygromycin by cell sorting after staining with the anti-Tspan5 (TS5-2) or anti-Tspan15 (5D4, IgG1) mAbs using a FACS Aria cell sorter (Becton Dickinson).

PC3 cells lacking Tspan5, Tspan15, or ADAM10, as well as U2OS cells lacking Tspan5 were generated using CRISPR/Cas9 gene editing technology. For ADAM10, we used a previously published sequence (TGCTCCTCTCCTGGGCGGCG; [Seegar et al, 2017]) and the target sequences for Tspan5 (TCTTCCATCACCGATCTCGG) and Tspan15 (CAGAAAAAGTTCAAGTGCTG) were selected using the CRISPR design tool available at the Broad Institute (https://portals.broadinstitute.org/gpp/public/analysis-tools/sgrna-design). The corresponding guide DNA sequences were cloned into the lentiCRISPRv2 plasmid (#52961; Addgene) according to the instructions of the Zhang laboratory (https://www.addgene.org/52961/) (Ran et al, 2013). The plasmids were transfected as above, and the cells were treated after 36–48 h with 5 $\mu$g/ml puromycin for 36–48 h. Cells negative for the antigen of interest were sorted as above.

### Flow cytometry analysis

Cells were detached with Trypsin–EDTA, washed in complete DMEM, and incubated for 1 h at 4°C with 10 $\mu$g/ml primary antibody or hybridoma supernatant. After three washings, the cells were incubated for 30 min at 4°C with an Alexa647-conjugated goat anti-mouse antibody. The cells were analyzed using an Accuri C6 flow-cytometer (Becton Dickinson).

### Analysis of Notch activity

This analysis was performed as previously described (Moretti et al, 2010): U2OS cells (transduced with Notch1) were seeded at the concentration of 20,000 cells/cm². The cells were transfected 24 h later with the Notch luciferase reporter and Renilla plasmids using FuGene HD (Promega). 24 h later, the cells were co-cultured with OP9 or OP9-DLL1 at 35,000 cells/cm². The activities of firefly and Renilla luciferases were determined using a dual luciferase reporter assay (Promega) according to the manufacturer's instructions.

### Immunoprecipitation and Western blotting

Immunoprecipitations were performed as previously described with minor modifications (Charrin et al, 2001; Arduise et al, 2008). For immunoprecipitation, the cells were lysed in a lysis buffer (30 mM Tris, pH 7.4, 150 mM NaCl, 1 mM O-phenanthroline, and protease inhibitors) supplemented with 1% detergent (Brij 97, digitonin, or Triton X-100). In some experiments, the ADAM10 inhibitor Gi254023X at 5 $\mu$M was added to the buffer. In the experiment shown in Fig 1, 1 mM $CaCl_2$ and $MgCl_2$ was added to the buffer for maximal

tetraspanin/tetraspanin interaction (Charrin et al, 2002). After 30-min incubation at 4°C, the insoluble material was removed by centrifugation at 10,000$g$ and the proteins were immunoprecipitated by adding 10 $\mu$l protein G Sepharose beads and either 1 $\mu$g mAb or 0.4 $\mu$l ascitic fluid to 400–800 $\mu$l of the lysate. The immunoprecipitated proteins were separated by SDS–polyacrylamide gel electrophoresis and transferred to a PVDF membrane (GE Healthcare). Western blotting on lysates was performed using appropriate primary and Alexa Fluor 680-labelled secondary antibodies. Western blotting on immunoprecipitations was performed using biotin-labelled antibodies and fluorescent streptavidin except for the anti-Tspan15 and the CD81 mAbs. All acquisitions were performed using the Odyssey Infrared Imaging System (LI-COR Biosciences).

### Analysis of protein palmitoylation

All reagents used for the analysis of Tspan5 and Tspan15 palmitoylation by click chemistry were obtained from Life technologies. U2OS cells expressing GFP-tagged Tspan5, Tspan15, or the mutants were incubated overnight with 50 $\mu$M click-it palmitic acid (15-azidopentadecanoic acid) in serum-free medium. After three washes, the cells were lysed in PBS containing 1% Triton X-100, and the proteins were immunoprecipitated using GFP-trap beads (ChromoTek). After three washing, the beads were resuspended in 15 $\mu$l of a solution containing 50 mM Tris HCl, pH 8 and 0.5% SDS. Biotinylation of the proteins labelled with the modified palmitic acid was performed by subsequent addition of 15 $\mu$l 40 $\mu$M biotin alkyne in click-it reaction buffer, 1, 5 $\mu$l CuSO$_4$, 1, 5 $\mu$l reaction buffer additive 1, and after 3 min 3 $\mu$l click-it additive 2. After 1 h at room temperature, 20 $\mu$l 3× Laemmli buffer was added to the reaction. The samples were analyzed by Western blot using Alexa680-labelled streptavidin.

### Immunofluorescence and confocal microscopy

The cells grown in complete medium were fixed for 15 min with 4% paraformaldehyde at room temperature, washed in PBS and then incubated for 15 min in 50 mM NH$_4$Cl in PBS. For surface labelling, the cells were then incubated directly for 1 h with 10 $\mu$g/ml of antibodies in PBS supplemented with 0.1% BSA at room temperature. For intracellular labelling, the cells were incubated with antibodies in the same buffer supplemented with 0.1% saponin. The binding of primary antibodies was revealed using appropriate secondary reagents. The cells were mounted in Prolong Gold (Thermo Fisher Scientific) supplemented with DAPI and examined with a Leica SP5 confocal microscope (63× objective, 1.4 numerical aperture, zoom 3 or 6).

### Endocytosis assay: confocal microscopy

The cells were incubated for 1 h at 37°C with the anti-ADAM10 mAb 11G2 coupled to DyLight 650. After fixation or cooling to 4°C, the surface pool is then labelled with an anti-mouse polyclonal antibody coupled to Alexa Fluor 568 or Alexa Fluor 488. Images were acquired every 0.25 $\mu$m throughout the height of the cells using a Leica SP5 confocal microscope (63× objective, 1.4 numerical aperture, airy = 1, zoom 3). Quantification was performed using a

protocol developed in Icy imaging software (de Chaumont et al, 2012) (http://icy.bioimageanalysis.org). The purpose of this protocol is to use the surface staining ("surface image") to generate a mask allowing removing the surface labelling from the total staining ("total image"). An inherent difficulty of this approach is that the results may vary considerably according to the threshold applied to the images. By definition, the internalized pool is not labelled by the secondary reagent or minimally in rare cases of partial permeabilization upon fixation. The surface and internalized pool can, therefore, be distinguished by the ratio of the two fluorescence intensities, independently of any threshold applied. Thus, one of the first steps in our protocol is to divide the "total image" by the "surface image." A threshold image is then generated, which allows removing from the total image the fraction of the signal that overlaps with the surface labelling, yielding a new image with only the intracellular labelling ("internal image"). The intracellular signal is quantified using the spot detector plugin. The total and surface images are quantified after applying a threshold determined using the K-mean method. In HeLa cells, the relative signals of the internalized fraction to the surface fraction were so different in the various transfectants that using the K-mean method on each cell type was not reliable. We, therefore, applied on all HeLa transfectants a threshold calculated on parental cells using the K-mean method. The percentage of endocytosed mAbs was obtained by dividing the value of the internal signal by that of the total signal. Three to eight fields were acquired for each experiment and averaged. At least three independent experiments were performed for each cell type. All images of internalization assays are maximum-intensity projections of the z-stacks.

In some experiments, the cells were incubated with non-labelled mAbs. After labelling, the surface pool was stained with an anti-mouse polyclonal antibody coupled to Alexa Fluor 568, and the cells were incubated with an anti-mouse polyclonal antibody coupled to another fluorochrome (Alexa Fluor 488 or 647) in the presence of saponin to reveal the internalized mAb. Quantification using Icy was performed as above, except that an endocytosis index was obtained by dividing the value of the internal image by that of the surface image.

### Endocytosis assay: flow cytometry

The cells were incubated at 37°C with 10 $\mu$g/ml DyLight 650–labelled ADAM10 mAb for different times. After detachment using accutase, the cells were incubated at 4°C with a PE-labelled secondary reagent, and the fluorescence was measured by flow-cytometry. In each experiment, a fraction of cells was not incubated with the ADAM10 mAb at 37°C, but at 4°C after detachment, and subsequently with the PE-labelled secondary reagent. This allows determining the ratio of PE (in FL2 channel) to DyLight 650 (in FL4 channel) fluorescence when the ADAM10 mAb is only at the cell surface, and thus an estimation of the surface fraction of ADAM10 from the DyLight 650 fluorescence in the cells incubated with the mAb at 37°C. The mean fluorescence intensity in the FL4 channel corresponds to the sum of the fluorescence of the mAb present at the cell surface and internalized. We estimated the fluorescence intensity corresponding to the internalized mAb (FL4$_i$) as follows: FL4$_i$ = [FL4n-(FL2$_n$ × FL4$_0$/FL2$_0$)], where FL2$_0$ or FL2$_n$ and FL4$_0$ or FL4$_n$

are the PE and DyLight 650 mean fluorescence intensity of non-treated cells (labelled after detachment) and cells treated for n hours. The ratio $FL4_i/FL4_0$ was then calculated to express the data relatively to the amount present at the cell surface at the steady state level.

## Statistical analysis

Statistical analysis was performed with GraphPad Prism on independent experiments as indicated in the legend to the figures. All graphs show the mean ± SEM.

# Supplementary Information

# Acknowledgements

This work was funded by core funding from the Institut National de la Santé et de la Recherche Médicale (INSERM) as well as by specific grants from NRB-Vaincre le Cancer, the Institut du Cancer et d'Immunogénétique, the Institut National du Cancer, the Agence Nationale de la Recherche and the comité "Essonne" of the Ligue Nationale Contre le Cancer. Etienne Eschenbrenner was a recipient of fellowships from the French Ministry of Research and from NRB-Vaincre le Cancer and Stéphanie Jouannet a recipient of a fellowship from NRB-Vaincre le Cancer.

## Author Contributions

E Eschenbrenner: formal analysis and investigation.
S Jouannet: formal analysis and investigation.
D Clay: formal analysis and investigation.
J Chaker: investigation.
C Boucheix: resources and writing—review and editing.
C Brou: resources and writing—review and editing.
MG Tomlinson: resources and writing—review and editing.
S Charrin: conceptualization, resources, formal analysis, investigation, and writing—review and editing.
E Rubinstein: conceptualization, formal analysis, supervision, funding acquisition, validation, investigation, methodology, and writing—original draft, review, and editing.

## Conflict of Interest Statement

The authors declare that they have no conflict of interest.

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
