## [Reviewer comments · Life Science Alliance]

Life Science Alliance

TspanC8 tetraspanins differentially regulate ADAM10 endocytosis and half-life.

Etienne Eschenbrenner, Stéphanie Jouannet, Denis Clay, Joëlle Chaker, Claude Boucheix, Christel Brou, Michael G. Tomlinson, Stéphanie Charrin, and Eric Rubinstein

DOI: <https://doi.org/10.26508/lsa.201900444>

Corresponding author(s): Eric Rubinstein, Inserm

Review Timeline:

Submission Date:	2019-05-31
Editorial Decision:	2019-07-09
Revision Received:	2019-11-12
Editorial Decision:	2019-11-15
Revision Received:	2019-11-20
Accepted:	2019-11-21

Scientific Editor: Andrea Leibfried

Transaction Report:

July 9, 2019

Re: Life Science Alliance manuscript #LSA-2019-00444-T

Dr. Eric Rubinstein
Inserm
Inserm, U1135
91 bd de l'hôpital
France 75013
France

Dear Dr. Rubinstein,

Thank you for submitting your manuscript entitled "TspanC8 tetraspanins differentially regulate ADAM10 endocytosis and half-life." to Life Science Alliance. The manuscript was assessed by expert reviewers, whose comments are appended to this letter.

As you will see, the reviewers appreciate your data and provide constructive input on how to further strengthen your manuscript. We would thus like to invite you to submit a revised version to us, addressing the reviewer comments. Importantly, the manuscript needs re-structuring (all reviewers), controls for expression levels should get provided (rev#2), and the functional data should get either removed (rev#3) or extended as suggested by rev#1.

Thank you for this interesting contribution to Life Science Alliance. We are looking forward to receiving your revised manuscript.

Sincerely,

B. MANUSCRIPT ORGANIZATION AND FORMATTING:

Reviewer #1 (Comments to the Authors (Required)):

Review of the manuscript entitled "TspanC8 tetraspanins differentially regulate ADAM10 endocytosis and half-life", submitted to Life-Science-Alliance by Eschenbrenner et al.

This manuscript addresses the different role of the C8 Tetraspanins Tspan5 and Tspan15 in regulating ADAM10 in different cell types. Overall, this is a high-quality study, in which gain- and loss of function experiments in different cell types are used to examine how these two TspanC8 proteins affect the levels and endocytosis and half-life of ADAM10. The authors use different approaches (FACS, immunofluorescence, biochemical studies such as Western blot and cell surface labeling, Co-IP) to uncover significant differences in how Tspan5 and Tspan15 affect ADAM10. The main conclusion is that Tspan5 promotes endocytosis of ADAM10, whereas Tspan15 leads to stabilization of ADAM10 on the cell surface. Structure/function analyses of chimera between Tspan5 and Tspan15 provides additional insights into the protein domains responsible for these differences. Moreover, the authors show that mutations in palmitoylation site in Tspan5 and 15 do not affect their role in ADAM10 transport or Notch processing. Finally, the authors demonstrate that an antibody against ADAM10 can lead to downregulation of ADAM10 in U2OS cells but not in PC3 cells, with a corresponding effect on the ability of ADAM10 to process Notch, APP and N-cadherin.

Overall, the data are of high quality and the interpretation is supported by the results. Given the important roles of ADAM10 and the TspanC8 proteins in development and disease, this careful characterization of the differential regulation of ADAM10 by Tspan5 versus Tspan15 provides important new information for the field that should also be of interest to scientists working in other related areas, such as intracellular sorting and regulation of proteases. The main critique is that the last two figures use functional assays for ADAM10 activity, whereas the previous figures are mainly concerned with the role of Tspan5 versus Tspan15 in regulating ADAM10 endocytosis. For completeness sake, it would be good to explore how the gain and loss of function of Tspan5 and Tspan15 affect the function of ADAM10 towards Notch, APP and N-cadherin processing. In addition, there are some very minor comments, as outlined below.

1) Regarding the functional studies shown in figure 9 and 10, it is not clear why the authors explore Notch signaling in the context of palmitoylation mutants in figure 9, and Notch, APP and N-cadherin processing in figure 10. It would be helpful to look at the same assays for the different mutants or experimental conditions, i.e. include APP and N-cadherin processing in figure 9. Moreover, since the effects of the 11G2 antibody in Figure 10 C are quite subtle, it would be good to also measure changes to the shed ectodomains of APP and N-cadherin.

2) Along these lines, it would be good to test how the CRISPR-deletion of Tspan5 and Tspan15 as well as the overexpression of these proteins affects the activity of ADAM10 towards APP and N-cadherin.

3) Minor comments:

a) It would be helpful if the order of the figures follows the order that they are first mentioned in the text (Fig. 2E and F is mentioned after figure 3).

b) On page 8, the last word should be Fig. 3D instead of C.

c) Page 18, middle of 2nd paragraph: The effect of the 11G2 antibody on Notch signaling could also simply be steric.

d) Please carefully go over the manuscript to remove minor typos, e.g. page 6, 2nd paragraph, "letter" should be "latter", or in the concluding remarks "is a bad reflect" should be something like "is

a poor reflection", to provide two examples.

Reviewer #2 (Comments to the Authors (Required)):

This paper describes the complex relationship between the tetraspanin members Tspan5 and Tspan15, and their client protein ADAM10, a cell surface shedding protease. This is an important topic because ADAM10 controls the cleavage of several very important cell surface proteins, and thereby, signalling. The main conclusion I draw from this paper is that it's complex: it does not really answer many of the questions that one would want to understand, but it does provide extensive data that will be of significance to the field. The simple one line proposal is that Tspan15 stabilises ADAM10 at the cell surface whereas Tspan5 can, at least in some contexts, accelerate ADAM10 endocytosis.

My overriding comment about the paper is that the data could be presented in a way that makes it easier to understand. The use of different cell types and different methodologies, while justified in the text, is scattered throughout the results in a way that, at least to me, made it hard to build a clear argument for the conclusion. I am not sure how, but I feel that there might be a more systematic way of presenting the data that makes it easier for the reader.

Other points (in no particular order)

Expression levels of different constructs may be critical when over-expressing genes. I think this needs to be addressed systematically. The data should exist already so I don't think this is a very onerous request.

There is quite a lot of speculation about the reasons for results that don't easily fit the trends. I think these should be left for the discussion unless they are essential for the narrative of the results. Aim to simplify the results section.

I am confused about why Tspan5 shows very little effect in some experiments (eg Fig 2 EF, but others, too). Can they explain this better, given their conclusion that Tspan5 accelerates endocytosis?

Given that they use multiple cell types in different experiments, perhaps they can comment on whether they believe that there are physiologically significant cell/tissue differences, or whether the differences they are see are idiosyncrasies of tissue culture cell lines.

Minor point: they should probably refer to the 'plasma' membrane rather than 'cell' membrane, which is a vague term that could mean anything.

Reviewer #3 (Comments to the Authors (Required)):

Review of Rubinstein MS

This manuscript reports the differential effects of two tetraspanin C8 proteins, tspan5 and tspan15, on ADAM10, a key protease that plays roles in Notch signal transduction and APP cleavage at the

alpha secretase site. The authors investigate how the two tspan affect ADAM10 surface levels, endocytic rates, and cellular activity using Notch signaling or substrate cleavage in several well-established cancer cell lines. They observe that tspan5 accelerates ADAM10 endocytosis, whereas tspan15 suppresses it. The difference in activity appears to depend, at least in part, on differences in sequence between the C-terminal tails of the tspan proteins, but swapping the tails is not sufficient to swap the activities, suggesting that other regions of the tspan also plays a role in the differential trafficking effect.

Overall, the authors provide sufficient evidence to support their claim that tspan5 promotes endocytosis relative to tspan15. The functional data in Figures 8 (other than the left panels of B and D, which confirm that domain swapping of the LEL does not meaningfully affect the endocytic phenotype) and 10, however, show very weak overall differences in signal, and it is not clear whether the magnitude of these observed effects is physiologically relevant. Instead, these figures detract from the (much stronger) trafficking data, and it is this reviewer's opinion that the paper would be greatly strengthened by removing these figures and refocusing the manuscript exclusively on the endocytic findings instead.

Some minor points:

- 1) Some figure callouts are off: p8: Fig 3C (bottom) should be 3D; p9: Fig. 5E (doesn't exist) should be 4E, I believe.
- 2) The writing is often difficult to follow (experimental design and logic). Figures 1-7 might benefit occasionally from inclusion of the experimental scheme, especially for endocytosis assays, etc...

Response to Reviewers

Reviewer #1 (Comments to the Authors (Required)):

This manuscript addresses the different role of the C8 Tetraspanins Tspan5 and Tspan15 in regulating ADAM10 in different cell types. Overall, this is a high-quality study, in which gain- and loss of function experiments in different cell types are used to examine how these two TspanC8 proteins affect the levels and endocytosis and half-life of ADAM10. The authors use different approaches (FACS, immunofluorescence, biochemical studies such as Western blot and cell surface labeling, Co-IP) to uncover significant differences in how Tspan5 and Tspan15 affect ADAM10. The main conclusion is that Tspan5 promotes endocytosis of ADAM10, whereas Tspan15 leads to stabilization of ADAM10 on the cell surface. Structure/function analyses of chimera between Tspan5 and Tspan15 provides additional insights into the protein domains responsible for these differences. Moreover, the authors show that mutations in palmitoylation sites in Tspan5 and 15 do not affect their role in ADAM10 transport or Notch processing. Finally, the authors demonstrate that an antibody against ADAM10 can lead to downregulation of ADAM10 in U2OS cells but not in PC3 cells, with a corresponding effect on the ability of ADAM10 to process Notch, APP and N-cadherin.

Overall, the data are of high quality and the interpretation is supported by the results. Given the important roles of ADAM10 and the TspanC8 proteins in development and disease, this careful characterization of the differential regulation of ADAM10 by Tspan5 versus Tspan15 provides important new information for the field that should also be of interest to scientists working in other related areas, such as intracellular sorting and regulation of proteases. The main critique is that the last two figures use functional assays for ADAM10 activity, whereas the previous figures are mainly concerned with the role of Tspan5 versus Tspan15 in regulating ADAM10 endocytosis. For completeness sake, it would be good to explore how the gain and loss of function of Tspan5 and Tspan15 affect the function of ADAM10 towards Notch, APP and N-cadherin processing. In addition, there are some very minor comments, as outlined below.

1) Regarding the functional studies shown in figure 9 and 10, it is not clear why the authors explore Notch signaling in the context of palmitoylation mutants in figure 9, and Notch, APP and N-cadherin processing in figure 10. It would be helpful to look at the same assays for the different mutants or experimental conditions, i.e. include APP and N-cadherin processing in figure 9. Moreover, since the effects of the 11G2 antibody in Figure 10 C are quite subtle, it would be good to also measure changes to the shed ectodomains of APP and N-cadherin.

Because both reviewers 1 and 3 find the effect of anti-ADAM10 mAb on APP and E-cadherin cleavage quite weak, we have removed these data from the manuscript, allowing us to examine only one functional test, namely Notch signaling.

2) Along these lines, it would be good to test how the CRISPR-deletion of Tspan5 and Tspan15 as well as the overexpression of these proteins affects the activity of ADAM10 towards APP and N-cadherin.

We have previously reported the effect of overexpression and down-regulation of Tspan5 and Tspan15 on the cleavage of APP and N-cadherin (Jouannet et al., CMLS, 2016).

3) Minor comments:

a) It would be helpful if the order of the figures follows the order that they are first mentioned in the text (Fig. 2E and F is mentioned after figure 3).

We have restructured the manuscript to make the flow of ideas more logical and to follow the order of the figures in the text.

b) On page 8, the last word should be Fig. 3D instead of C.

Corrected

c) Page 18, middle of 2nd paragraph: The effect of the 11G2 antibody on Notch signaling could also simply be steric.

This part has been removed

d) Please carefully go over the manuscript to remove minor typos, e.g. page 6, 2nd paragraph, "letter" should be "latter", or in the concluding remarks "is a bad reflect" should be something like "is a poor reflection", to provide two examples.

Corrected

Reviewer #2 (Comments to the Authors (Required)):

This paper describes the complex relationship between the tetraspanin members Tspan5 and Tspan15, and their client protein ADAM10, a cell surface shedding protease. This is an important topic because ADAM10 controls the cleavage of several very important cell surface proteins, and thereby, signalling. The main conclusion I draw from this paper is that it's complex: it does not really answer many of the questions that one would want to understand, but it does provide extensive data that will be of significance to the field. The simple one line proposal is that Tspan15 stabilises ADAM10 at the cell surface whereas Tspan5 can, at least in some contexts, accelerate ADAM10 endocytosis.

My overriding comment about the paper is that the data could be presented in a way that makes it easier to understand. The use of different cell types and different methodologies, while justified in the text, is scattered throughout the results in a way that, at least to me, made it hard to build a clear argument for the conclusion. I am not sure how, but I feel that there might be a more systematic way of presenting the data that makes it easier for the reader.

We have restructured the manuscript to make the flow of ideas more logical and to follow the order of the figures in the text. We hope it is now easier to understand.

Other points (in no particular order)

Expression levels of different constructs may be critical when over-expressing genes. I think this needs to be addressed systematically. The data should exist already so I don't think this is a very onerous request.

We now show the control of expression levels by Flow-cytometry in the new supplementary Fig 4.

There is quite a lot of speculation about the reasons for results that don't easily fit the trends. I think these should be left for the discussion unless they are essential for the narrative of the results. Aim to simplify the results section.

The re-structuration of the text takes this comment into account.

I am confused about why Tspan5 shows very little effect in some experiments (eg Fig 2 EF, but others, too). Can they explain this better, given their conclusion that Tspan5 accelerates endocytosis?

We don't know why Tspan5 shows no effect in some experiments. We have added a comment on this in the discussion page 16.

Given that they use multiple cell types in different experiments, perhaps they can comment on whether they believe that there are physiologically significant cell/tissue differences, or whether the differences they are see are idiosyncrasies of tissue culture cell lines.

Because the strong positive effect of Tspan15 on ADAM10 levels was observed with both gain and loss of function, using all assays, and because Tspan15 KO mice show a decrease in ADAM10 expression levels in the brain, we believe that this observation is physiologically significant. We have now commented this point in the discussion page 16.

Minor point: they should probably refer to the 'plasma' membrane rather than 'cell' membrane, which is a vague term that could mean anything.

Done

Reviewer #3 (Comments to the Authors (Required)):

This manuscript reports the differential effects of two tetraspanin C8 proteins, tspan5 and tspan15, on ADAM10, a key protease that plays roles in Notch signal transduction and APP cleavage at the alpha secretase site. The authors investigate how the two tspan affect ADAM10 surface levels, endocytic rates, and cellular activity using Notch signaling or substrate cleavage in several well-established cancer cell lines. They observe that tspan5 accelerates ADAM10 endocytosis, whereas tspan15 suppresses it. The difference in activity appears to depend, at least in part, on differences in sequence between the C-terminal tails of the tspan proteins, but swapping the tails is not sufficient to swap the activities, suggesting that other regions of the tspan also plays a role in the differential trafficking effect.

Overall, the authors provide sufficient evidence to support their claim that tspan5 promotes endocytosis relative to tspan15. The functional data in Figures 8 (other than the left panels of B and D, which confirm that domain swapping of the LEL does not meaningfully affect the endocytic phenotype) and 10, however, show very weak overall differences in signal, and it is not clear whether the magnitude of these observed effects is physiologically relevant. Instead, these figures detract from the (much stronger) trafficking data, and it is this reviewer's opinion that the paper would be greatly strengthened by removing these figures and refocusing the manuscript exclusively on the endocytic findings instead.

The Fig. 10 showed that an antibody that diminishes ADAM10 surface levels reduces to some extent the cleavage of APP and N-cadherin. We agree that the effect is mild, and have therefore removed these data from the manuscript.

Fig. 8 analyzes the molecular determinants responsible for the differential effects of Tspan5 and Tspan15 on ADAM10 trafficking and ADAM10-dependent Notch signaling. Since Tspan5 and Tspan15 have an opposite effect on both endocytosis and Notch signaling, it is logical to

hypothesize that the latter might be the consequence of the former, and thus to compare the influence of different chimeras on both aspects. As noted by the reviewer, the data show that the LEL does not contribute to the differences, and this is new information. But we believe that there is much more valuable information in this figure.

To test the impact of the chimeras on ADAM10 function, we chose to focus on Notch signaling, on the one hand because of the key physiological role of ADAM10 in this important pathway and on the other hand because the protocol allows precise quantification, and the effect of Tspan15 expression is strong (>60% inhibition) and highly reproducible (see the error bars, and note that we reproduce with an untagged Tspan15 construct the effect of the GFP-tagged construct (Jouannet et al., CMLS 2016)).

We agree with the reviewer that there were only weak differences in the right panels of the original fig. 8. But this has to be compared to the effect of Tspan5 or Tspan15 in the left panels. In Fig. 8D, the fact that all chimeras stimulate an intermediate level of ADAM10 surface expression level indicates that different regions of Tspan5 or Tspan15 contribute to their action on ADAM10.

Of special interest is the finding that replacing either the N- or the C-ter of Tspan15 by that of Tspan5 abolished its ability to inhibit Notch signaling (Fig. 8C). We believe that this finding provides an additional argument in favor of the hypothesis that Tspan15 might not inhibit the intrinsic ability of ADAM10 to play a role in Notch signaling, but may instead localize ADAM10 into a membrane compartment where it cannot play a role in Notch signaling.

Of note, we have improved the data in Figure 8: 1- We have analyzed two new chimeras in which the cytoplasmic N tails of Tspan5 and Tspan15 are exchanged. 2- By increasing the signal to noise ratio in the endocytosis assays (Fig. 8B), we could observe differences between several chimeras that were not observed in our previous set of experiments.

Some minor points:

1) Some figure callouts are off: p8: Fig 3C (bottom) should be 3D; p9: Fig. 5E (doesn't exist) should be 4E, I believe.

corrected

2) The writing is often difficult to follow (experimental design and logic). Figures 1-7 might benefit occasionally from inclusion of the experimental scheme, especially for endocytosis assays, etc...

We have re-structured the manuscript and we hope it will be easier to follow. We have included an experimental scheme for most confocal analyzes.

November 15, 2019

RE: Life Science Alliance Manuscript #LSA-2019-00444-TR

Dr. Eric Rubinstein
Inserm
Inserm, U935
91 bd de l'hôpital
France 75013
France

Dear Dr. Rubinstein,

Thank you for submitting your revised manuscript entitled "TspanC8 tetraspanins differentially regulate ADAM10 endocytosis and half-life. I assessed the revised version and appreciate how you responded to the reviewers' concerns. I would thus be happy to accept your manuscript for publication here, pending minor revision to meet our formatting guidelines:

- Please mention all figure panels for Fig 6 in the legend
- Please mention the statistical test used in the figure legend when mentioning p-values.
- The light grey labeling (Tspan5+ etc) in figure 1 is difficult to see, please increase contrast
- I would like to suggest to replace GAM with secondary antibody in Fig 3-5 and suppl figures
- Please define LEL within the manuscript text as Large extracellular loop upon first mentioning; please do so as well for PE

A. FINAL FILES:

B. MANUSCRIPT ORGANIZATION AND FORMATTING:

Sincerely,

November 21, 2019

RE: Life Science Alliance Manuscript #LSA-2019-00444-TRR

Dr. Eric Rubinstein
Inserm
Inserm, U935
14 Av Paul Vaillant Couturier
Villejuif 94800
France

Dear Dr. Rubinstein,

Thank you for submitting your Research Article entitled "TspanC8 tetraspanins differentially regulate ADAM10 endocytosis and half-life." It is a pleasure to let you know that your manuscript is now accepted for publication in Life Science Alliance. Congratulations on this interesting work.

DISTRIBUTION OF MATERIALS:

Again, congratulations on a very nice paper. I hope you found the review process to be constructive and are pleased with how the manuscript was handled editorially. We look forward to future exciting submissions from your lab.

Sincerely,
